# Identifying monitoring information needs that support the management of fish in large rivers

Timothy D. Counihan[1]*, Kristen L. Bouska[2], Shannon K. Brewer[3], Robert B. Jacobson[4], Andrew F. Casper[5], Colin G. Chapman[6], Ian R. Waite[7], Kenneth R. Sheehan[8], Mark Pyron[9], Elise R. Irwin[3], Karen Riva-Murray[10], Alexa J. McKerrow[11], Jennifer M. Bayer[12]

1 U.S. Geological Survey, Western Fisheries Research Center, Columbia River Research Laboratory, Cook, Washington, United States of America, 2 U.S. Geological Survey, Upper Midwest Environmental Sciences Center, La Crosse, Wisconsin, United States of America, 3 U.S. Geological Survey, Alabama Cooperative Fish and Wildlife Research Unit, Auburn, Alabama, United States of America, 4 U.S. Geological Survey, Columbia Environmental Research Center, Columbia, Missouri, United States of America, 5 Illinois Natural History Survey, Illinois River Biological Station, Havana, Illinois, United States of America, 6 Oregon Department of Fish and Wildlife, Ocean Salmon and Columbia River Program, Clackamas, Oregon, United States of America, 7 U.S. Geological Survey, Oregon Water Science Center, Portland, Oregon, United States of America, 8 U.S. Geological Survey, Grand Canyon Monitoring and Research Center, Flagstaff, Arizona, United States of America, 9 Ball State University, Muncie, Indiana, United States of America, 10 U. S. Geological Survey, Northeast Region, Troy, New York, United States of America, 11 U.S. Geological Survey, Science Analytics and Synthesis, Core Science Systems, Raleigh, North Carolina, United States of America, 12 U.S. Geological Survey, Northwest-Pacific Islands Region, Cook, Washington, United States of America

* tcounihan@usgs.gov

**Data Availability Statement:** Relevant data are within the paper and its Supporting Information files. Data used in this study are available from: Fig S1 - https://waterdata.usgs.gov/nwis/uv/?site_no=

## Abstract

Management actions intended to benefit fish in large rivers can directly or indirectly affect multiple ecosystem components. Without consideration of the effects of management on non-target ecosystem components, unintended consequences may limit management efficacy. Monitoring can help clarify the effects of management actions, including on non-target ecosystem components, but only if data are collected to characterize key ecosystem processes that could affect the outcome. Scientists from across the U.S. convened to develop a conceptual model that would help identify monitoring information needed to better understand how natural and anthropogenic factors affect large river fishes. We applied the conceptual model to case studies in four large U.S. rivers. The application of the conceptual model indicates the model is flexible and relevant to large rivers in different geographic settings and with different management challenges. By visualizing how natural and anthropogenic drivers directly or indirectly affect cascading ecosystem tiers, our model identified critical information gaps and uncertainties that, if resolved, could inform how to best meet management objectives. Despite large differences in the physical and ecological contexts of the river systems, the case studies also demonstrated substantial commonalities in the data needed to better understand how stressors affect fish in these systems. For example, in most systems information on river discharge and water temperature were needed and available. Conversely, information regarding trophic relationships and the habitat requirements of larval fishes were generally lacking. This result suggests that there is a need to better

07228000 Fig S2 - https://waterdata.usgs.gov/nwis/dv/?site_no=09380000 Fig S4 - https://waterdata.usgs.gov/nwis/dv/?site_no=14105700 Fig S6 - https://umesc.usgs.gov/data_library/fisheries/graphical/fish_front.html, https://umesc.usgs.gov/data_library/fisheries/fish1_query.shtml Additional information regarding accessing these datasets are included in the supporting figure captions.

**Funding:** This work was funded in part by U.S. Geological Survey's Core Science Systems Mission Area. This research also was conducted using in-kind contributions of the Ball State University, Illinois Natural History Survey, Oregon Department of Fish and Wildlife, Pacific Northwest Aquatic Monitoring Partnership, the Oklahoma and Alabama Cooperative Fish and Wildlife Research Units, and the U.S. Geological Survey Columbia Environmental Research Center, Grand Canyon Monitoring and Research Center, Oregon Water Science Center, Upper Midwest Environmental Sciences Center, and Western Fisheries Research Center.

**Competing interests:** The authors have declared that no competing interests exist.

understand a set of common factors across large-river systems. We provide a stepwise procedure to facilitate the application of our conceptual model to other river systems and management goals.

## Introduction

Long-term monitoring has benefited a variety of freshwater ecosystems, including large rivers like the Ohio [1, 2] and Illinois rivers [3]. Large-river systems are complex, making the development of effective monitoring programs especially difficult. Large rivers are dynamic systems with high variability in spatio-temporal physicochemical characteristics and biotic assemblages [4]. The inherent complexity of large rivers makes biotic assemblages logistically difficult to sample [5] and the mechanisms of change difficult to understand. Large rivers represent the culmination of vast stream networks and, thus, integrate and accumulate the effects of multiple stressors at varying spatial scales [6]. The spatial and temporal complexity associated with large rivers has hindered the identification of mechanisms driving declining populations of aquatic species [7–10]. To exacerbate the complexity, large rivers commonly have within-channel structural alterations (e.g., dams, river training structures, [11]) and often exhibit legacy effects from historical land uses [12]. To deal with the complexity, some areas of aquatic science recommend monitoring be used to test the linkages developed first through conceptual models (e.g., environmental flows, [13–15]).

Conceptual models are useful tools to help guide the design of monitoring programs [16]. The identification of questions relevant to conservation and management efforts requires some foresight and knowledge of the complexities of the system being monitored. For example, it is generally well accepted that the native range of the federally-listed Arkansas River Shiner (*Notropis girardi)* is truncated [17], though there is uncertainty surrounding the multiple threats affecting the species [18]. Reducing the uncertainty associated with the decline of the Arkansas River Shiner through the implementation of a hypothesis-driven monitoring program would facilitate confidence in moving forward with a recovery plan. This is where conceptual models are quite useful; they can serve as the foundation to guide hypothesis-driven monitoring programs [14, 16] and identify key ecosystem processes and factors that may directly or indirectly affect management outcomes [19–22].

Understanding factors affecting the status and trends of fishes is of interest to multiple stakeholder groups across multiple jurisdictions. Fishes provide economic benefits to businesses that serve recreational interests, commercial and recreational fishers, tribal members for whom fish are an integral part of their cultural identity [23], and to local and state governments who derive revenue from these activities. Fish populations are affected by the integration of physical habitat, water quality, environmental contamination, habitat fragmentation, and overall ecosystem productivity [24–27]. Consequently, fish are often the focus of management and monitoring programs (e.g., [28]). However, because fish integrate the effects of so many components of the ecosystem, the success of efforts to manage fishes can be affected by unintended consequences of mitigation on factors not directly targeted by the actions. Without consideration of the effects of management on non-target ecosystem components, unintended consequences may limit management efficacy.

Our goal is to demonstrate how a structured, yet flexible, conceptual model (CM) can be used to identify the types of monitoring information needed to understand the range of factors affecting large-river fishes. Our CM includes a hierarchically structured conceptualization of

ecosystem characteristics based on CMs originally developed by Harwell, Myers [29] and elaborated by Jacobson and Berkley [30]. We chose to incorporate the tiered conceptualization of ecosystem characteristics proposed by Jacobson and Berkley 30] in part because it allows users to define their own biotic or abiotic interests. In this paper, we discuss the structure and development of the CM. We apply the CM to case studies to illustrate the flexibility and applicability of this approach and use it to identify monitoring information needs specific to disparate management goals. More specifically, for each case study, we use the CM to hypothesize how human activities affect fish populations and then identify information needs required to evaluate the hypothesized relationships. We then posit the spatial and temporal scales of the management goal addressed in the conceptual model, inferences needed to inform the management goal, and data collection requirements needed to make the inferences.

## Conceptual model

### Overview of approach

Since 2012, scientists working on large rivers across the United States have participated in a U. S. Geological Survey (USGS) led forum intended to improve our understanding of large-river ecosystems. The collaborative forum has worked to identify best practices of long-term monitoring programs [31] and evaluate trends in fish assemblages across rivers [32]. As this group of scientists moved toward linking changes in fish populations and assemblages to human activities, there was a need to develop a process to help identify and prioritize the information needed to assess trends in large river fishes. To that end, the group of scientists that comprise the collaborative forum, and that are the authors of this manuscript, participated in a workshop that was convened in Hood River, Oregon in May 2017, to jointly adapt, apply, and qualitatively evaluate a conceptual model for developing hypotheses that detail stressors affecting fishes arising from natural and anthropogenic sources [33].

Prior to the workshop, the experts that comprise the collaborative forum were asked to identify important anthropogenic activities that affect fish populations or communities in the river basins they work in (Fig 1). The information from this exercise was summarized and the anthropogenic activities were grouped into driver categories (Fig 2). A general form of the CM was composed. The summarized information describing anthropogenic activities affecting fish and the general form of the CM were then given to the experts prior to the workshop.

During the workshop we discussed and refined the CM form (Fig 3). We then had representatives from each river system represented at the workshop choose a management goal to address. Then, through a facilitated discussion led primarily by the representative of the river system being addressed, we 1) elaborated tiered conceptualizations of ecosystem characteristics to reflect the large-river systems and management goals being examined, 2) used knowledge of the fish species' life history and population bottlenecks to relate biological ecosystem characteristics to habitat requirements, 3) hypothesized pathways describing how anthropogenic and natural drivers affect large-river fish populations either indirectly (e.g., effects on flow regime, habitat, trophic resources, etc.) or directly (e.g., competition with invasive species), and 4) hypothesized interactions within ecosystem characteristic tiers that could affect the management goal. Based on this exercise, we chose four case studies to refine for use in this manuscript (Fig 1).

After the workshop, we held a series of conference calls with workshop participants with expertise in the selected case studies to refine all aspects of the CMs and associated information. During the calls, we started with the CM from the workshop and discussed and clarified CM components, pathways, and inter-tier interactions. We then characterized whether, based on the expert knowledge of workshop participants, there was a strong, moderate, or weak

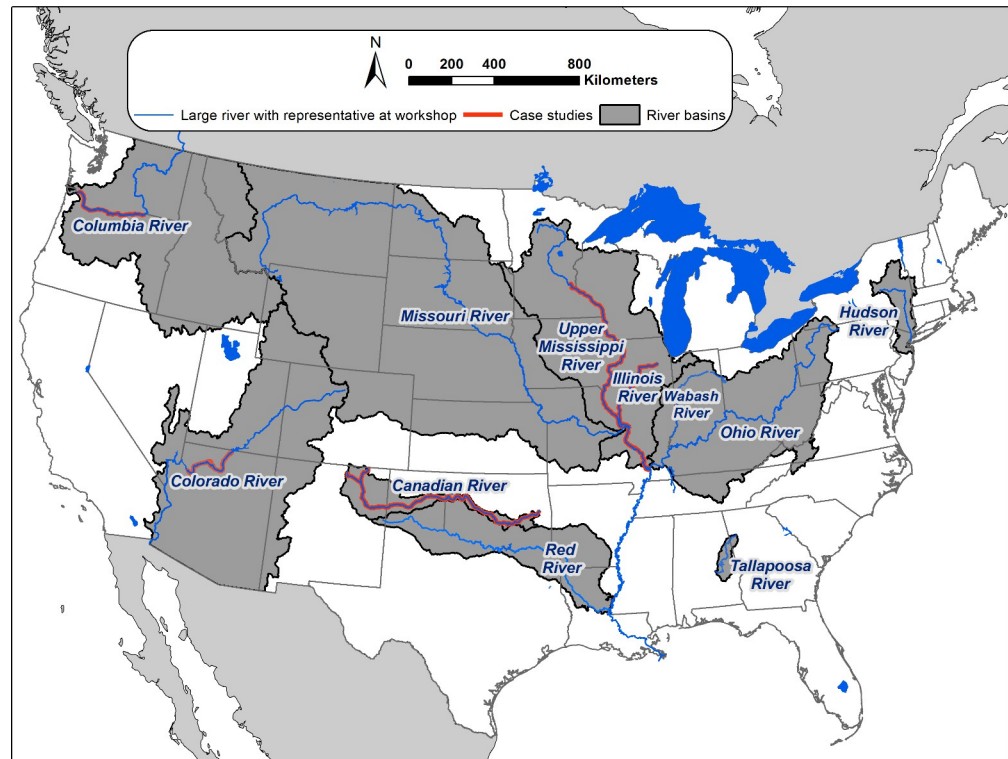

**Fig 1. Map of rivers and watersheds represented by scientists that convened to develop a conceptual model that depicts how natural and anthropogenic drivers interact with habitats, biological systems, and fish in large rivers.** River segments where we conducted case studies that applied the conceptual model to identify monitoring information needs associated with management goals are highlighted in red. Data used: 1:2,000,000-Scale Hydrologic Unit Boundaries–source: U.S. Geological Survey; U.S. States (Generalized), U.S. Rivers (Generalized) 2005, U.S. Lakes (Generalized) 2005, and World Countries 2005 –source: ESRI.

understanding of the pathways and interactions. A list of the information needed to understand the relationships described in the CM was developed. We then had the case study leader classify whether the data required to understand the information needs were available, insufficient, or not available. For information needs that were classified as insufficient or not available, we characterized the spatial and temporal scales at which data should be collected to make inferences that support the evaluation of management goals.

We then encouraged representatives from each basin to share the CM and the case study narrative with other experts familiar with the river system and management goal. This outreach took several forms including sharing the CM with working groups tasked with implementing the management goal, discussions with peers familiar with the management goal, and presenting the CM at regional conferences. The intent was to garner opinions from outside the workshop participants. If needed, the CMs incorporated the feedback received.

## Hierarchical structure of conceptual model

Our CM is a hierarchical conceptualization of how anthropogenic and natural drivers relate to multiple tiers representing the physical and biological components of large-river ecosystems (Fig 3). Natural drivers included in the CM were physiographic, climatic, and biogeographic factors that control fluxes of water, mass, energy, and genetic information in a watershed [30]. The physiographic factors, such as lithology, soils, and watershed topography, exert control on

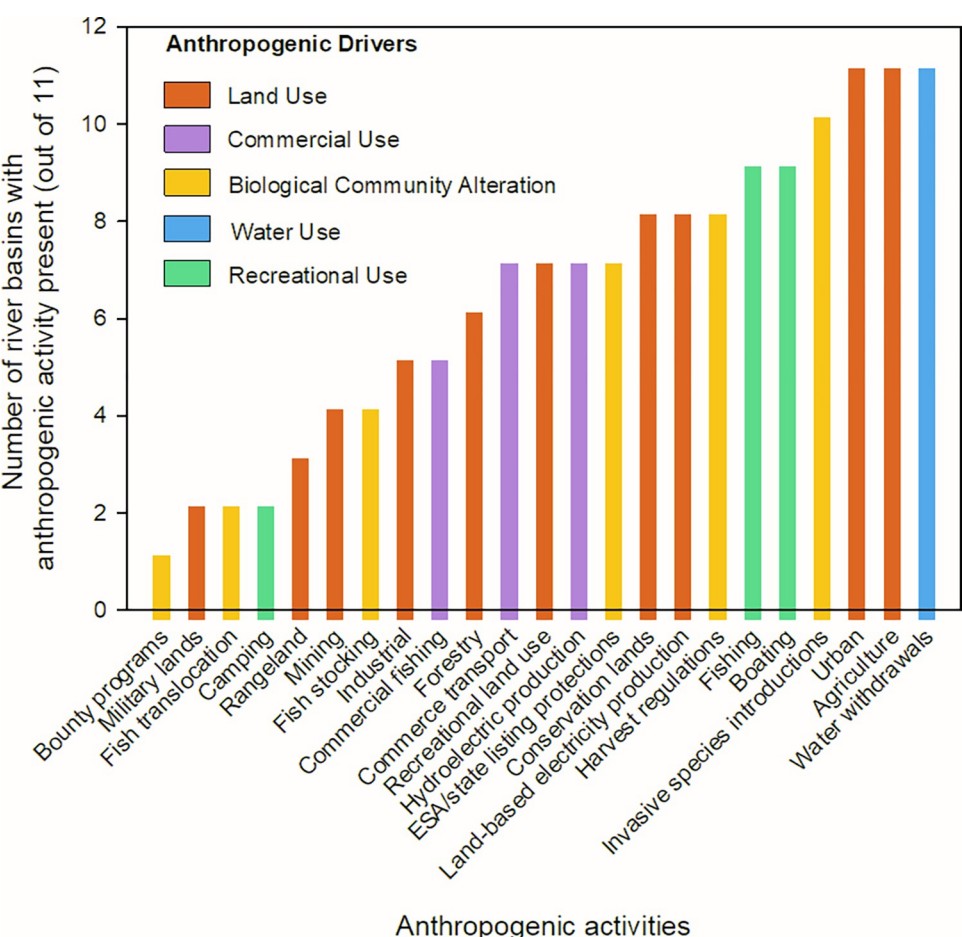

**Fig 2. The results of a query to scientists from the Canadian, Colorado, Columbia, Hudson, Illinois, Ohio, Missouri, Red, Tallapoosa, Upper Mississippi, and Wabash Rivers, U.S to identify anthropogenic activities that affect large-river fishes in the river systems they represent.** Anthropogenic activities were classified into five driver categories.

water, sediment, and geochemical fluxes (e.g., nutrients) into the river corridor. Physiography is generally static over time frames of decades to centuries. Climate controls fluxes of atmospheric energy and moisture into the watershed. Unlike physiography, climate is more likely to vary over relatively shorter temporal scales. Biogeography describes the native organism assemblage in the watershed (e.g., [34]) and the natural flux of genetic information due to immigrations, emigrations, mutations, and extinctions. Changes arising from the biogeography driver includes altered spatial distribution of organisms within the watershed, which, in turn, may alter the effects of natural system regimes on the river corridor. For example, natural variation of the type and distribution of vegetation can affect the time series and magnitude of runoff events [30].

We created five categories of anthropogenic drivers to characterize a range of human activities that affect large rivers: land use, commercial use, biological community alteration, water use, and recreation (Fig 2). The land-use category is intended to reflect different ways humans use landscape resources that affect large rivers. We defined commercial use as the use of river resources for marketable enterprises that did not involve water removal or transfer. We included biological community alteration to represent the intentional or non-intentional

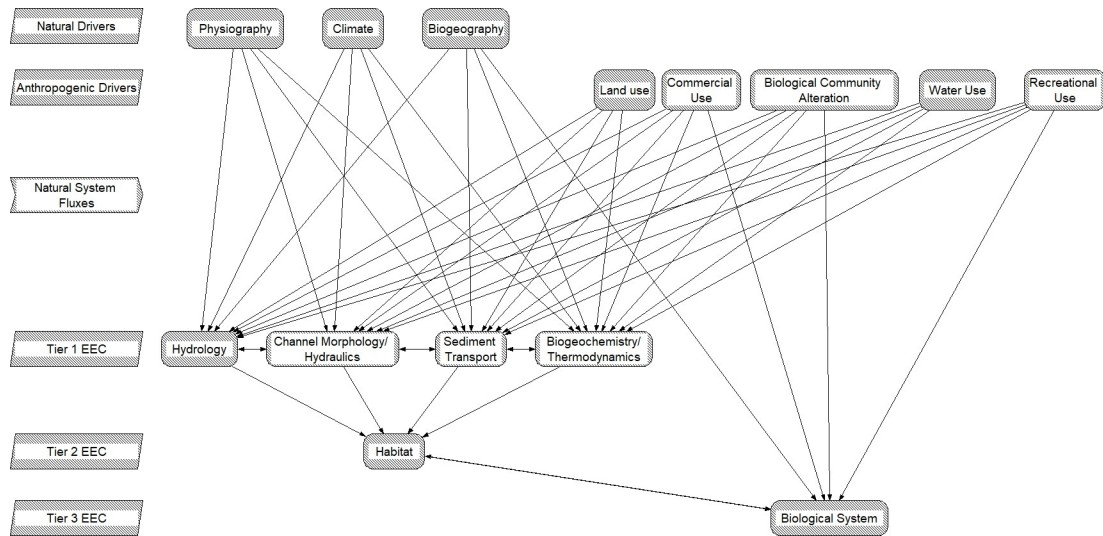

**Fig 3. Tiered hierarchical conceptualization of how anthropogenic and natural drivers relate to physical and biological components of large-river ecosystems.** Essential ecosystem characteristics (EECs) are groupings of ecosystem components. Tier 1 EECs represent physical and chemical effects; fundamental measures of process that are directly affected by anthropogenic and natural drivers. Tier 2 EECs represent a broad habitat category that is intended to encompass the physical, chemical, and biological components of the riverine habitats that influence reproduction, growth, and survival of biotic communities. The Tier 3 EEC represent components of the biological systems that respond to changes in the hierarchical components of the conceptual model.

human alteration or manipulation of the river's biological community (e.g., introductions of non-native fish). Recreational use was defined as the use of river resources for leisure activities (e.g., fishing, boating). We considered water use a direct commercial or non-commercial use of river water that involved the removal or transfer of water.

Our CM includes hierarchically structured essential ecosystem characteristics (EEC), originally developed by Harwell, Myers [29] and described in detail by Jacobson and Berkley (30). Briefly, EECs are characteristics that can be classified into similar groups based on the way they link to biological endpoints [30]. Tier 1 EECs are measurable characteristics that describe processes that can significantly alter the morphological or chemical characteristics within a river channel. The Tier 1 categories we considered were 1) hydrology, 2) channel morphology/ hydraulics, 3) sediment transport, and 4) biogeochemistry/thermodynamics. Tier 2 EECs are broadly described as physicochemical or biological components of "habitat" that are hypothesized to affect fish populations or assemblages (e.g., growth, survival, reproduction, [35]). Lastly, Tier 3 EECs represent components of the hypothesized biological system that are affected by the cascading (e.g., degradation of egg quality caused by increases in sediment deposition) or direct (e.g., predation by invasive species) effects of anthropogenic and natural drivers. Tier 2 characteristics are particularly important because these are the factors that can be examined at scales most often sampled by fisheries managers [36]. The specific components that comprise Tier 2 and 3 EECs are flexible and can be adapted and elaborated depending on the river system and specific management goal being addressed.

We retained aspects of the approach taken by Jacobson and Berkley [30]] with respect to how our model represents interactions between drivers and EECs, but with key differences. Since we were interested in representing how human activities affect large river ecosystems, our approach acknowledges that anthropogenic and natural drivers interact and alter the expected characteristics of Tier 1 EECs. Similar to Jacobson and Berkley [30], our model depicts a stress associated with a natural or anthropogenic driver to Tier 1 EECs as fluxes in

natural system regimes that alter the frequency, magnitude, duration, timing, or rate of change in natural systems or by the imposition of a hard-structural constraint on channel form. The natural system regimes considered in our CM were hydraulic, hydrologic, sediment, temperature, light, and biogeochemistry. Graphically, the natural system fluxes were represented by arrows connecting anthropogenic and natural drivers to Tier 1 EECs. Similarly, hypothesized pathways between EECs, that depict the expression of the cascading effects of anthropogenic and natural drivers, and interaction within EECs were depicted as arrows. For example, fragmentation of river systems resulting from altered hydrologic and/or hydraulic regimes caused by dams, weirs, levees, and other factors are frequently cited sources of stress to large-river fishes [37]. Fragmentation can prevent fish from migrating and/or dispersing through their natural reproductive ranges and from accessing critical habitats [38]. To depict a scenario where the presence of a dam is altering hydrologic and/or hydraulic regimes resulting in habitat fragmentation, the CM would 1) show an arrow from an anthropogenic stress (i.e., dam as a commercial activity) to a Tier 1 EEC (e.g., channel morphology/hydraulics) that would depict a natural system flux (e.g., altered hydrologic and/or hydraulic regime) that would then 2) manifest as a stress caused by habitat fragmentation depicted by an arrow between the Tier 1 EEC and a Tier 2 EEC (e.g. habitat) that would then 3) manifest as an effect on a Tier 3 component, shown by an arrow between Tier 2 and Tier 3. All stress pathways and interactions were classified with respect to the strength of understanding of the relationships based on expert opinion.

## Spatial and temporal context

The successful characterization of how human activities influence large-river fishes is dependent upon integrated concepts of scale. Fish distributions in rivers can vary spatially within river basins in relation to naturally occurring and human induced landscape characteristics [32]. Fish distributions can also vary seasonally, annually, and over longer times in response to changing environmental conditions [39]. Consequently, the spatial and temporal scope of fish management goals often varies within and between large-river systems and agencies. For data collected by monitoring programs to have the highest relevance, the spatial and temporal scales appropriate for scientific investigation and management must also be time and geographic-context specific. For example, the management of White Sturgeon (*Acipenser transmontanus*) in the Columbia River varies by reservoir or river segment and season [40, 41]. The effects of hydropower development on White Sturgeon vary spatially and temporally as well, so the spatial and temporal context of the data needed to understand the effects needs to be considered. For instance, hydropower peaking operations, that can vary by dam and season, affect river discharge in a river reach on a diel and even hourly basis [42], whereas water storage and other management actions can affect seasonal discharges over a broader geographic scale [40]. Understanding the spatial and temporal context needed to inform management will help ensure relevant information is collected.

We considered spatial and temporal resolution in our CMs. We defined spatial extent as: local network–synonymous with Hydrologic Unit Code (HUC) 2 basins [43, 44]; segment–the portion of a river between two major tributary confluences [45] or other hydrogeomorphic features [46]; reach—the length of river occurring between breaks in channel slope caused by man-made dams or other hydrogeomorphic features [45]; patch–an area used by an organism (e.g., for reproduction or resource attainment) that can vary both spatially and temporally depending on the species of interest [47, 48]. For our purposes, the spatial scales considered are hierarchically nested such that segments occur within local networks, reaches occur within segments, and patches occur at the sub-reach scale. Temporal units considered were daily,

seasonal, annual, and decadal. The temporal units were used to denote both the scale of inferences needed to support the management goal and the scale at which data should be collected to inform the inferences.

## Case studies

We applied the CM to four case studies. For each, we followed the pathways of stress from Tier 1 EECs to the biological endpoint associated with the management goal to identify information needs. We then characterized the spatial and temporal scales of the management goal, the scientific inferences needed to inform the management goal, and that data collection needs to occur to support the inferences for monitoring information needs identified as requiring additional data for each of the case studies. To summarize similarities across case studies, we generalized the stressors and inter- tier interactions identified in the case studies and then summarize the similarities by EEC tier. More context for the river systems characterized in the case studies can be found in S1 Appendix.

## South Canadian River

Native populations of the federally-threatened Arkansas River Shiner are believed to be restricted to two fragmented portions of the South Canadian River [49]. The Arkansas River Shiner is hypothesized to be affected by several anthropogenic activities that primarily affect water quality and quantity (Fig 4). Three reservoirs on the South Canadian River have altered discharge patterns (S1 Fig), and fragmented river habitats. Two known native populations of Arkansas River Shiner occupy the two remaining river segments of sufficient length and complexity to allow eggs to drift the time required to successfully complete their early life history. Small impoundments for agriculture use, road crossings, groundwater pumping and other local water extractions (e.g., oil and gas) threaten to further fragment existing habitat. Fragmentation could also be problematic for upstream fish migrations; there is some evidence that Arkansas River Shiners migrate upstream to spawn to achieve adequate drift distances for their offspring [50]. It has also been speculated that this species might benefit from access to floodplain habitats [51], but we are unaware of efforts to examine that hypothesis. Changes in the flow patterns may also relate to the expansion of salt cedar *Tamarix* spp. and other non-native riparian species that constrain the channel and inhibit channel habitat complexity [52, 53]. Changes to the riparian corridor can also alter the availability of drifting invertebrates for Arkansas River Shiner feeding (i.e., Coleoptera, Hymenoptera; [54]). Channel complexity acts to slow the transport of eggs [49] and may prevent eggs from being washed into downstream reservoirs where survival is hypothesized to be extremely low. Climate change is expected to increase the intensity and frequency of drought events within this region [55, 56], which may exacerbate habitat fragmentation, promote all-terrain vehicle traffic within the river channel causing direct mortality on stranded fish (Gene Wilde, Texas Tech University, Personal Comm.), and concentrate contaminants and salinity [57]. The tolerances of Arkansas River Shiner to salinity concentrations and many other contaminants are unknown (see S1 Table; [18]). Lastly, introductions of non-native fishes have occurred within the basin. The primary concern is the presence of Red River Shiner (*Notropis bairdi*) because it is suspected to reproduce in a similar manner and be a possible competitor to the Arkansas River Shiner [57].

The results of the CM exercise that characterized factors affecting the Arkansas River Shiner in the South Canadian River suggested the critical life-history bottlenecks for the Arkansas River Shiner are successful spawning and recruitment to the first year. Impediments that limit our understanding of factors that lead to successful spawning and recruitment included the effects of channel morphology and hydraulics on the quality and quantity of larval rearing

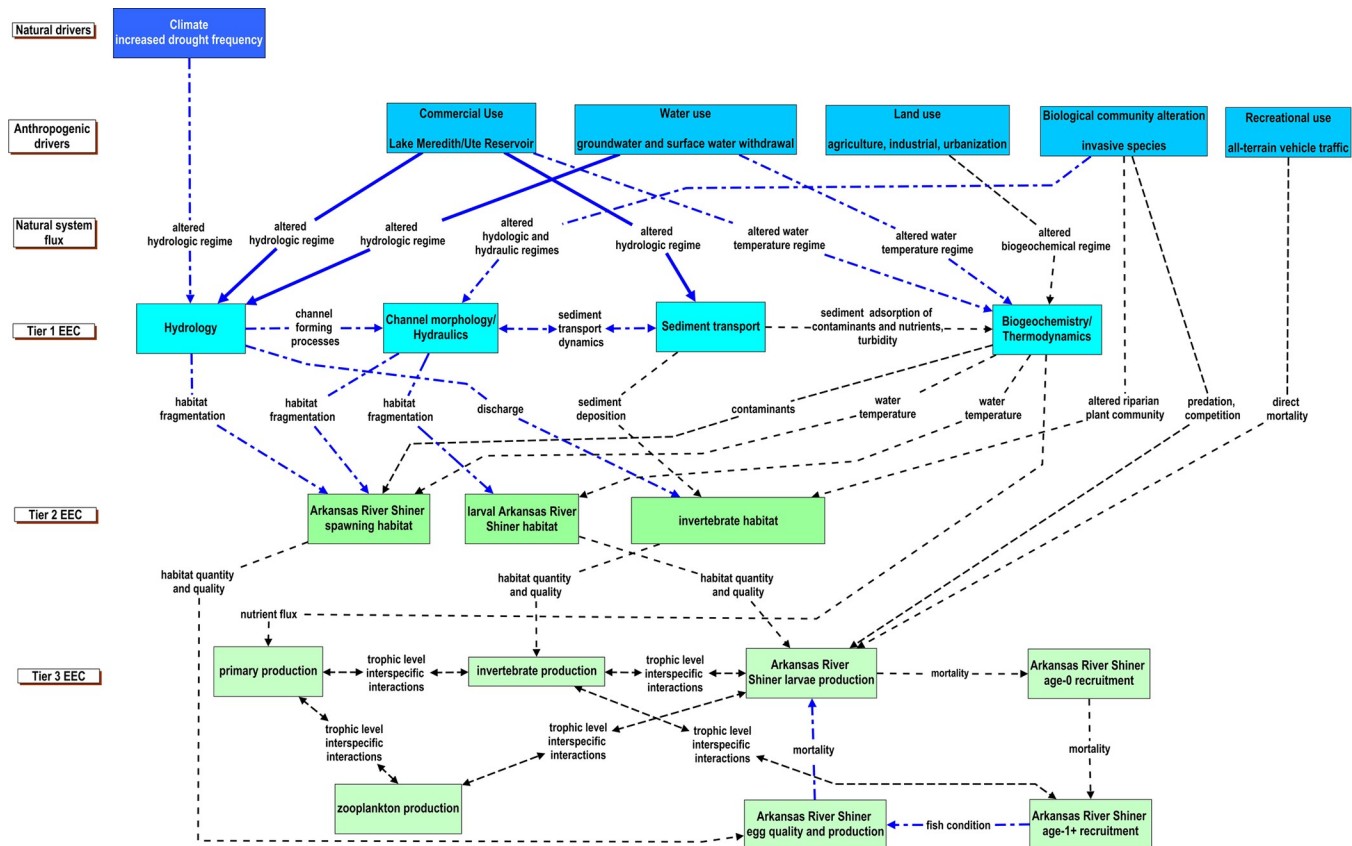

**Fig 4. Conceptual model describing the relationship of natural and anthropogenic drivers to essential ecosystem characteristics (EECs) affecting the recruitment of the Arkansas River Shiner in the South Canadian River in New Mexico, Texas, and Oklahoma.** Essential ecosystem characteristics are groupings of ecosystem components. Tier 1 EECs represent physical and chemical effects; fundamental measures of process that are directly affected by anthropogenic and natural drivers. Tier 2 EECs represent a broad habitat category that is intended to encompass the physical, chemical, and biological components of the riverine habitats that influence reproduction, growth, and survival of biotic communities. The Tier 3 EECs represent components of the biological systems that respond to changes in the hierarchical components of the conceptual model. The strength of our understanding of how natural and anthropogenic drivers interact with habitats, biological systems, and fish in large rivers is represented by the different types of lines in the figure. Solid blue lines depict a strong understanding of the relationship, the dotted-dashed blue line represents a moderate understanding of the relationship, and the black dashed line represents a weak understanding of the relationship. The different types of lines also represent the strength of our understanding of within EEC-tier relationships.

habitat, and subsequent effects on larval production (S1 Table). Water use and other drivers occurring at relatively coarse spatial and temporal scales are the hypothesized drivers related to degradation of reproductive habitat for the Arkansas River Shiner (Fig 5). A temporal lag in responses at finer scales (i.e., improved habitat) would be anticipated with management actions at these coarser spatial scales (e.g., water releases from dams); though, providing connectivity via minimal water releases would occur relatively quickly. Although there are gages on the South Canadian River, the spacing of the gages is not sufficient to have a full understanding of flow patterns between the gages given the semi-arid nature of the basin and potential for reaches to be affected by water withdrawals such as groundwater pumping. Our understanding of the species life history is well established; however, the effects of human pressures on the species and associated habitat has not been well studied (i.e., production, survival). As stressors propagate through Tier 1 to Tier 2 and Tier 3, the level of uncertainty increased such that it is not possible to define a preferred hypothesis for Arkansas River Shiner recruitment failure. The status of information needed to understand the hypothesized stress pathways and interactions was mostly characterized as insufficient or not available (S1 Table).

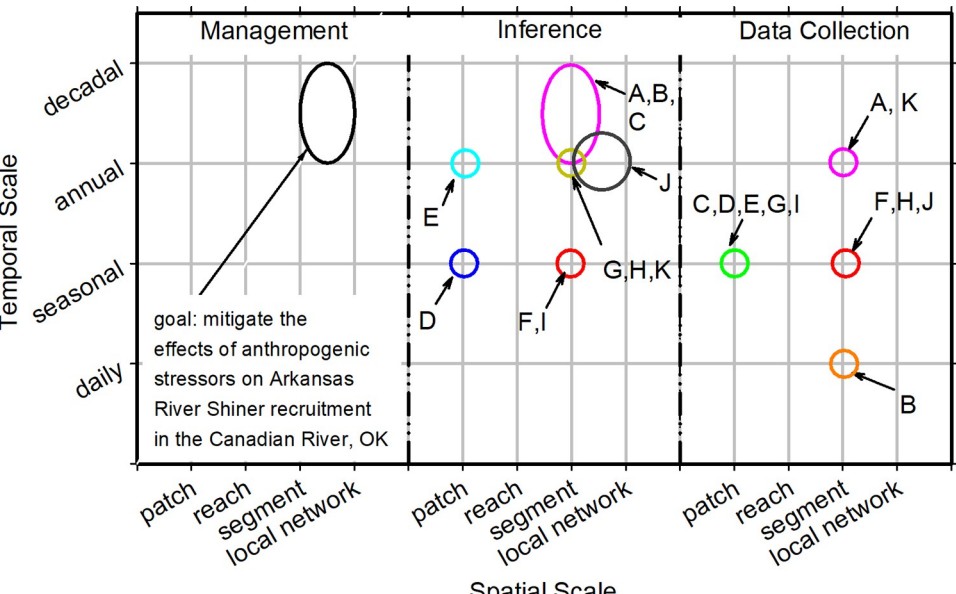

**Fig 5. The spatial and temporal scales of the management goal, the scientific inferences needed to inform the management goal, and that data collection needs to occur to support the inferences for monitoring information needs identified as requiring additional data in the case study addressing the recruitment of the Arkansas River Shiner in the Canadian River, Oklahoma (see S1 Table for additional detail).** A:Tier 1 EEC = channel morphology/ hydraulics; Stressor = altered hydraulic regime; B:Tier 1 EEC = biogeochemistry/thermodynamics; Stressor = altered water temperature regime; C:Tier 1 EEC = biogeochemistry/thermodynamics; Stressor = altered biogeochemical regime; D:Tier 2 EEC = Arkansas River Shiner spawning habitat; Stressors = contaminants, water temperature, habitat fragmentation; E:Tier 2 EEC = larval Arkansas River Shiner habitat, Stressors = water temperature, habitat fragmentation and Tier 2 EEC = invertebrate habitat, Stressors = altered riparian plant community, discharge, sediment deposition; F:Tier 3 EEC = primary production, Stressor = nutrient flux; G:Tier 3 EEC = invertebrate production; Stressor = invertebrate habitat quantity and quality and Tier 3 EEC = Arkansas River Shiner larvae production, Stressor = predation by invasive species; H:Tier 1 EEC = biogeochemistry/thermodynamics; Inter-tier interaction = Sediment adsorption of contaminants and nutrients and Tier 3 EEC = Arkansas River Shiner larvae production, Stressors = Arkansas River Shiner larvae habitat quantity and quality and Tier 3 EEC = Arkansas River Shiner age-0 recruitment; Inter-tier interaction = Arkansas River Shiner larvae mortality and Tier 3 EEC = Arkansas River Shiner age-1+ recruitment, Inter-tier interaction = Arkansas River Shiner age-0 mortality and Tier 3 EEC = all, Inter-tier interaction = trophic level interactions; I:Arkansas River Shiner larvae production; Stressor = direct mortality from recreational use (i.e., all-terrain vehicle and in-river traffic); J:Tier 3 EEC = Arkansas River Shiner egg quality and production; Stressor = Arkansas River Shiner spawning habitat quantity and quality; K:Tier 3 EEC = Arkansas River Shiner larvae production; Inter-tier interaction = Arkansas River Shiner egg mortality.

## Colorado River

The Humpback Chub (*Gila cypha*), a fish native to the Colorado River, was listed as endangered by the U.S. Fish and Wildlife Service in 1967 and given full protection under the Endangered Species Act of 1973 (ESA). To mitigate the effects of anthropogenic changes to the river on Humpback Chub, an understanding of the mechanisms by which Glen Canyon Dam and non-native species affect Humpback Chub is needed. A critical life-history bottleneck for Humpback Chub is recruitment into the first year class (Fig 6). Temperature, light, and seasonal high river discharge from snowmelt are thought to cue spawning behavior [58]. Hydropower development has dampened the range of river discharges of the Lower Colorado River within Grand Canyon. Historically, river discharge varied between 15 and 3400 m³/s, however discharge was greater in large flood events; current dam operations limit flows to a range of 140 to 1000 m³/s (S2 Fig). Resulting changes in turbidity and water temperature create risks to endangered Humpback Chub, and other endemic fish. For example, the quantity and quality

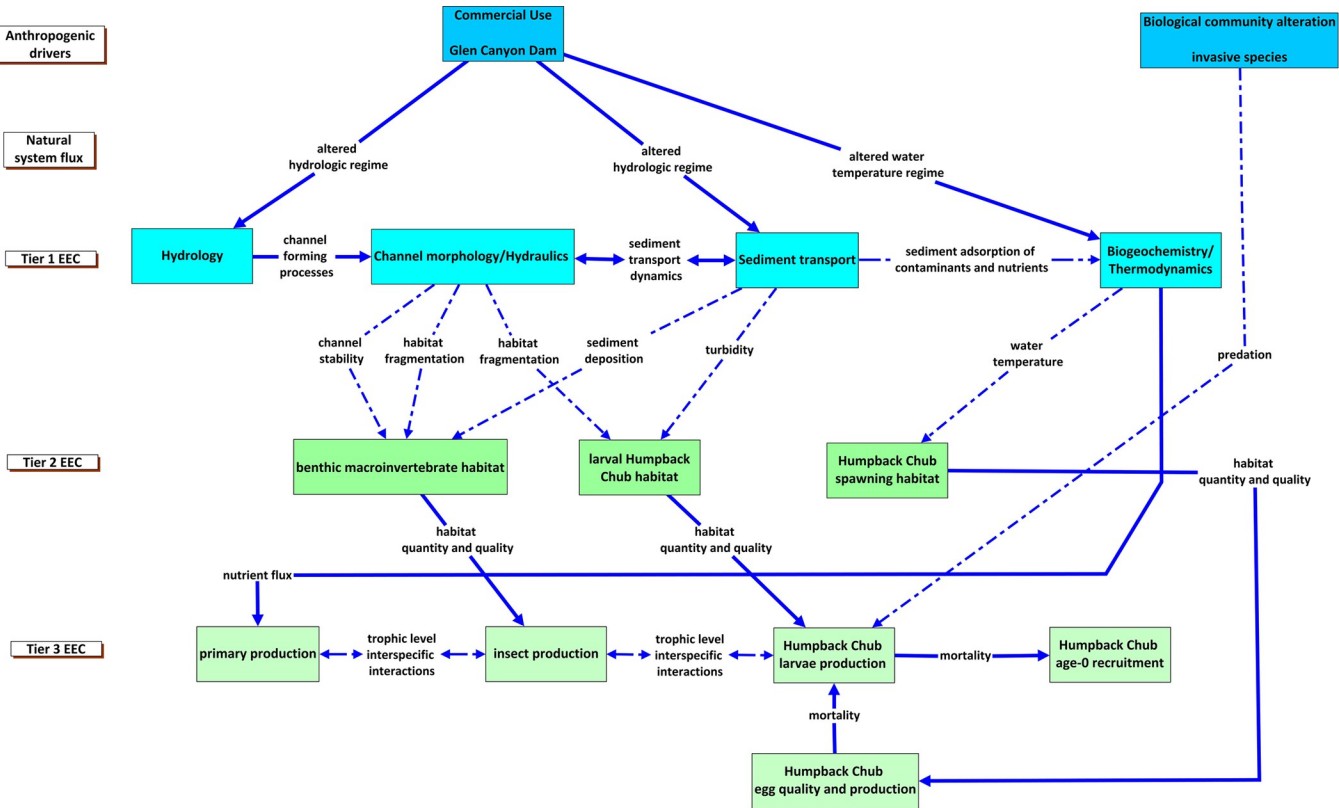

**Fig 6. Conceptual model describing the relationship of anthropogenic drivers to essential ecosystem characteristics (EECs) affecting the recruitment of Humpback Chub in the Colorado River between Glen Canyon Dam and Lake Mead, Arizona.** Essential ecosystem characteristics (EECs) are groupings of ecosystem components. Tier 1 EECs represent physical and chemical effects; fundamental measures of process that are directly affected by anthropogenic and natural drivers. Tier 2 EECs represent a broad habitat category that is intended to encompass the physical, chemical, and biological components of the riverine habitats that influence reproduction, growth, and survival of biotic communities. The Tier 3 EECs represent components of the biological systems that respond to changes in the hierarchical components of the conceptual model. The strength of our understanding of how natural and anthropogenic drivers interact with habitats, biological systems, and fish in large rivers is represented by the different types of lines in the figure. Solid blue lines depict a strong understanding of the relationship, the dotted-dashed blue line represents a moderate understanding of the relationship, and the black dashed line represents a weak understanding of the relationship. The different types of lines also represent the strength of our understanding of within EEC-tier relationships.

of habitat is reduced through changes in turbidity, biogeochemistry, and the temperature regime. Hypolimnetic water releases from Lake Powell maintain cold temperatures in the Colorado River downstream from Glen Canyon Dam; currently, spawning is limited to a single tributary, the Little Colorado River. As embryos survive into the larval stage, nursery habitats to support growth and foraging are essential [59–61]. A secondary risk to juvenile survival post-larval stage is predation by non-native species including Rainbow Trout (*Oncorhynchus mykiss*) and Channel Catfish (*Ictalurus punctatus*) [62]. Temperatures for well over 100 km downstream of Lake Powell are excellent for non-native, cold water species, including a closely managed world-class Rainbow Trout fishery at Lees Ferry. Rainbow and Brown Trout (*Salma trutta*) are currently managed as an invasive species downstream of the confluence of the Colorado and Little Colorado River, approximately 97 km downstream of Glen Canyon Dam, to mitigate predation upon native fishes including the endangered Humpback Chub.

The CM exercise documented a high understanding of the relationships of anthropogenic drivers to Tier 1 EECs, cascading to multiple, less-understood hypotheses about how these factors would combine to affect habitats at Tier 2 (Fig 6). High confidence in the linkages from

Tier 2 invertebrate habitat to Tier 3 insect production is followed by a lesser understanding of how insect production is linked to larval production. The uncertainty of the relations between food resources stands in contrast to high certainty that was ascribed to the linkages from larval chub habitat to larval chub production, and from spawning habitat to larval chub production and then to recruitment. Since 1997, the Glen Canyon Dam Adaptive Management Program has supported extensive monitoring and research across the spatial and temporal landscape of the Colorado River. As a result, the information needed to characterize some of the stressors is readily available (S2 Table). However, the status of some existing information was characterized as insufficient or not available. For the information needs characterized as being insufficient or not available, we identified the spatial and temporal scales at which data collection would facilitate the inferences needed to inform the management goal (S3 Fig). Understanding how these knowledge gaps affect Humpback Chub recruitment could identify strategies that will help achieve the management goal of improving Humpback Chub recruitment in the Colorado River.

## Columbia River

White Sturgeon is the largest freshwater fish in North America [63]. Like other sturgeon species, anthropogenic stressors have negatively affected White Sturgeon productivity. Our knowledge of factors affecting White Sturgeon productivity remain poorly understood [40]. Therefore, we used the CM to identify knowledge gaps associated with the hypotheses that dam construction and operation, land-use practices, and invasive species, in some combination, affect the recruitment of age-0 White Sturgeon (Fig 7). Within the basin, development of hydroelectric and water-storage dams have changed the magnitude and seasonality of the natural river discharge (S4 Fig) and thermal regimes [64], reduced the quantity and quality of spawning habitats [65, 66], and disrupted historical migration patterns [67]. Prior to hydropower development, White Sturgeon experienced a hydrograph that peaked during June-July due to snowmelt [64]. However, from 1949 to 1993 the average discharge in June decreased from 14,000 m$^3$/s to 6,000 m$^3$/s and the maximum water temperature has increased by 1.8˚C [64]. White Sturgeon likely used the natural hydrograph and thermal regime as cues to seek out optimal spawning habitats and initiate spawning [65, 68].

Factors other than river discharge and water temperature may also be affecting age-0 White Sturgeon recruitment [40]. In areas of the Columbia and Snake Rivers with hydropower development, White Sturgeon populations are functionally isolated by dams. Consequently, White Sturgeon depend on conditions within restricted reaches to sustain production. In some reaches, suitable rearing habitat exists, and individual growth rates are high, but spawning habitat is limited and recruitment of fish is poor [65]. In other reaches, favorable spawning conditions exist but growth of young fish may be density limited [69]. How the availability of food resources for larval and juvenile White Sturgeon varies among reservoirs may affect age-0 White Sturgeon recruitment. Research has also suggested that contaminants may affect White Sturgeon reproductive biology [70]. The introduction of non-native fishes has clearly affected the native fish assemblage in the Columbia River [32, 71]. Channel Catfish, Smallmouth Bass *Micropterus dolomieu*, and Walleye (*Sander vitreus*) that have been introduced into the Columbia River have all been shown to prey upon or compete with native fish species [71, 72] and may also affect White Sturgeon.

The CM (Fig 7) provides structure to the multiple competing hypotheses and indicates how anthropogenic drivers may be affecting Tier 1, 2, and 3 EEC's. Not surprisingly, data were available and the strength of understanding was high for relationships between white sturgeon life stages that are routinely sampled (e.g., adults) and metrics derived from data readily

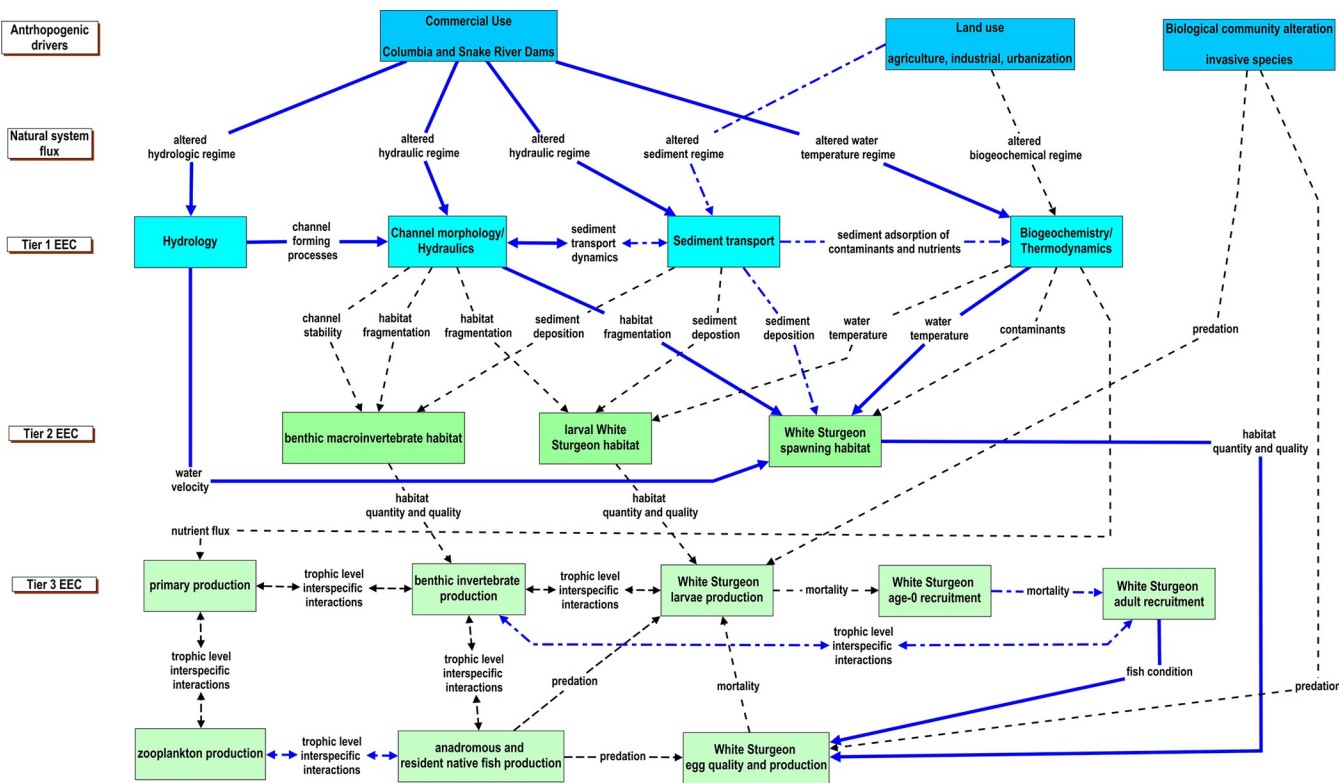

**Fig 7. Conceptual model describing the relationship of anthropogenic drivers to essential ecosystem characteristics (EECs) affecting the recruitment of White Sturgeon in the Columbia River, U.S. Essential ecosystem characteristics (EECs) are groupings of ecosystem components.** Tier 1 EECs represent physical and chemical effects; fundamental measures of process that are directly affected by anthropogenic and natural drivers. Tier 2 EECs represent a broad habitat category that is intended to encompass the physical, chemical, and biological components of the riverine habitats that influence reproduction, growth, and survival of biotic communities. The Tier 3 EECs represent components of the biological systems that respond to changes in the hierarchical components of the conceptual model. The strength of our understanding of how natural and anthropogenic drivers interact with habitats, biological systems, and fish in large rivers is represented by the different types of lines in the figure. Solid blue lines depict a strong understanding of the relationship, the dotted-dashed blue line represents a moderate understanding of the relationship, and the black dashed line represents a weak understanding of the relationship. The different types of lines also represent the strength of our understanding of within EEC-tier relationships.

collected at hydropower facilities (e.g., estimates of White Sturgeon spawning habitat derived from discharge and water temperature; S3 Table). However, for relationships between harder to sample White Sturgeon life stages (e.g., larvae), biota that require expertise and equipment atypical of traditional fisheries assessments in large rivers (e.g., benthic macroinvertebrates), or stressors that are described by metrics that require specialized modeling expertise (e.g., sediment transport dynamics), the existing information was insufficient or not available. For example, we identified the need to better understand the effects of channel morphology and hydraulics on benthic macroinvertebrate habitat, invertebrate production, and subsequent larval White Sturgeon production. The most certain pathways connected changes in hydrology, hydraulics, and temperature regimes to reduced spawning habitat in Tier 2, then to decreased egg quality and production at Tier 3.

Our results suggest there are stressors that can affect the management goal of increasing age-0 White Sturgeon recruitment that are poorly understood and that could confound efforts to manage White Sturgeon in the Columbia River. Our characterization of the spatial and temporal scales that data should be collected at could help guide future efforts to fill data gaps to support the inferences needed to address the goal of improving recruitment of age-0 White Sturgeon (S5 Fig).

## Upper Mississippi and Illinois Rivers

In the agricultural Midwest, basin-wide land uses affect the delivery of sediments, nutrients, and runoff to the Upper Mississippi and Illinois rivers [73–76]. Within the floodplain of these two large rivers, agriculture and residential land uses often rely upon the use of levees to isolate productive or developed lands during seasonal high-flow events. Within the channel, these rivers support commercial navigation with locks and dams and river-training structures, which have dramatically altered channel morphology and hydraulics throughout the system. Together, the cumulative effects of these modifications to the basin, floodplain, and river have implications for habitat diversity and native fish biodiversity [77]. Additionally, recent invasion and expansion of non-native species, namely Silver Carp (*Hypophthalmichthys molitrix*) and Bighead Carp (*H. nobilis)*, have direct and indirect effects on native fishes that likely compound or confound stress pathways on native fish biodiversity [78, 79]. Therefore, we used the CM to explore how these primary anthropogenic drivers have likely influenced fish habitats and associated life stages (Fig 8).

Increased sediment loads in combination with altered hydraulics and morphology have resulted in high rates of sedimentation, homogeneity of depth, and loss of low-velocity, off-

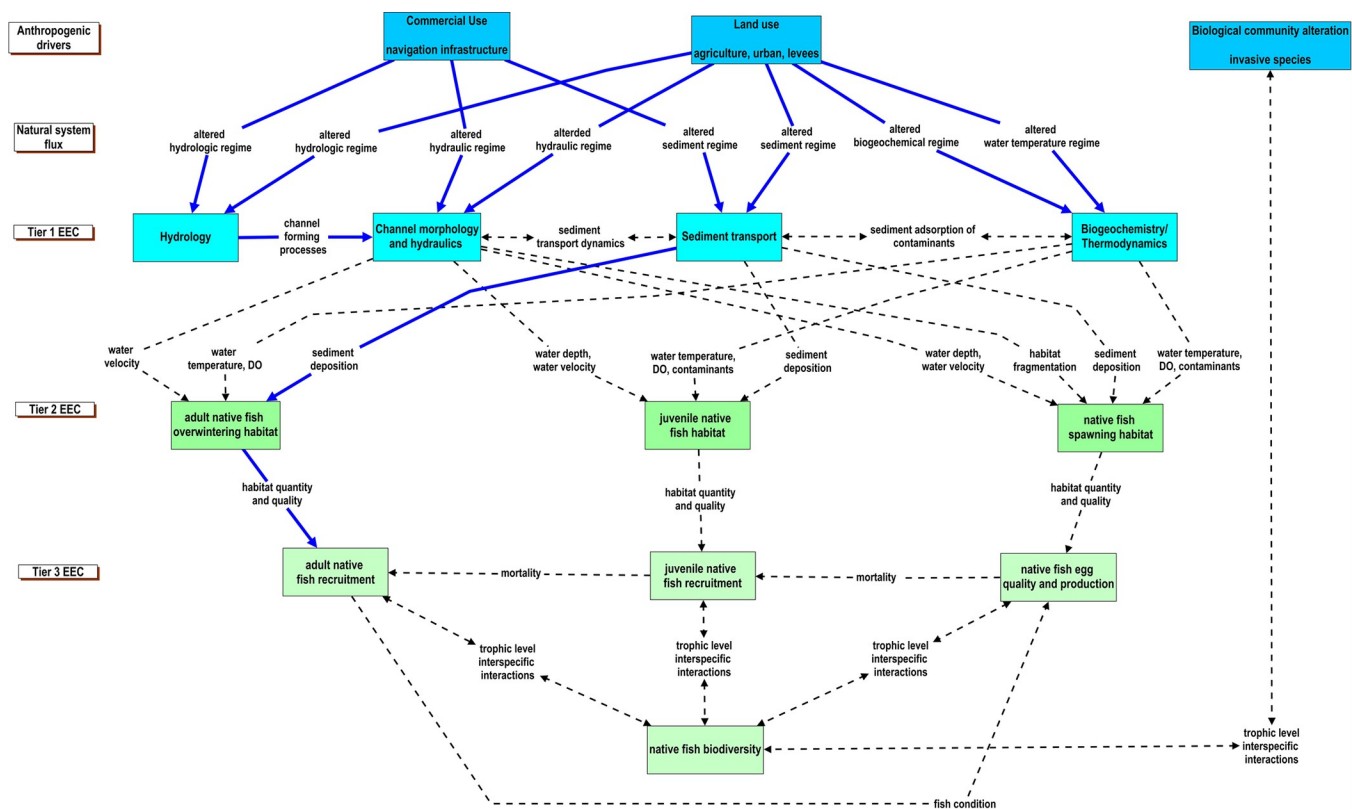

**Fig 8. Conceptual model of how anthropogenic drivers in the upper Mississippi and Illinois Rivers influence native fish habitats and recruitment.**
Essential ecosystem characteristics (EECs) are groupings of ecosystem components. Tier 1 EECs represent physical and chemical effects; fundamental measures of process that are directly affected by anthropogenic and natural drivers. Tier 2 EECs represent a broad habitat category that is intended to encompass the physical, chemical, and biological components of the riverine habitats that influence reproduction, growth, and survival of biotic communities. The Tier 3 EECs represent components of the biological systems that respond to changes in the hierarchical components of the conceptual model. The strength of our understanding of the relationships of how natural and anthropogenic drivers interact with habitats, biological systems, and fish in large rivers is represented by the different types of lines in the figure. Solid blue lines depict a strong understanding of the relationship, the dotted-dashed blue line represents a moderate understanding of the relationship, and the black dashed line represents a weak understanding of the relationship. The different types of lines also represent the strength of our understanding of within EEC-tier relationships.

channel areas [80, 81]. A diversity of off-channel habitat conditions (i.e., increased residence time, low velocity, warm temperatures, availability of food resources) support growth and development of larval and juvenile fishes [82, 83] and often provide important food resources for adult fishes [84–88]. Further, deep, low-velocity off-channel habitats are recognized as important refugia for a wide range of fishes during high-flow events and seasonal periods of low temperatures [89–92]. Loss of floodplain connectivity has eliminated the seasonal exchange of nutrients, organisms and organic matter between river and floodplain environments that support biological diversity and productivity [93, 94]. Reduced availability of spawning, nursery, foraging, or overwintering habitat conditions can serve as bottlenecks to fish populations through limited larval production, reduced growth, and increased overwinter mortality. For example, high sedimentation rates have been filling backwaters in the Illinois River for decades, thus limiting the availability of overwintering conditions for fishes that bioenergetically need a deep refuge with slow water velocities. Missing year-classes in this reach, represented by truncated size structure in the Largemouth Bass (*Micropterus salmoides*) population are hypothesized to be a result of periodic winter mortality (S6 Fig).

The application of our CM makes clear that while the general effects of anthropogenic drivers on hydrology, sediment transport, biogeochemistry and hydraulics and morphology are well understood, there is much less known about how those effects influence the quality and availability of required habitat conditions (Tier 2, Fig 8). Although there is likely overlap of habitat requirements among species with similar life histories, the diversity of habitat conditions necessary to support a native and diverse fish community has not been explored. Consequently, the existing information needed to assess the relationship between habitat quality and quantity, and egg production, juvenile recruitment, and adult survival of fish populations within the Upper Mississippi and Illinois rivers was categorized as insufficient to not available (Tier 3 Inter-tier interaction, Fig 8; S4 Table). Addressing these knowledge gaps could improve the effectiveness of habitat restoration efforts focused on maintaining a diverse native fish community. The spatial and temporal scales of data collection that would support needed inferences to address restoring and maintaining native fish biodiversity and habitat quantity and quality are characterized in S7 Fig.

## Similarities across case studies

We observed similarities in the stressors and interactions within EEC tiers across the four case studies. For Tier 1, an altered hydrologic regime was identified as a stressor to the hydrology EEC in all four rivers (Table 1). Presumably this is due to the ubiquitous effects of dams on the

**Table 1. Stressors or inter-tier interactions affecting Tier 1 Essential Ecosystem Characteristics (EEC) identified as an information need in the application of the conceptual model to case studies in the Canadian River (1), Colorado River (2), Columbia River (3), and Upper Mississippi and Illinois rivers (4).**

| Stressor or inter-tier interaction | Tier 1 EEC | | | |
|---|---|---|---|---|
| | Hydrology | Channel Morphology/Hydraulics | Sediment Transport | Biogeochemistry/Thermodynamics |
| Altered Hydrologic Regime | 1, 2, 3, 4 | 1 | 1, 2 | |
| Altered Hydraulic Regime | | 1, 3, 4 | 3 | |
| Altered Sediment Regime | | | 3, 4 | |
| Altered Water Temperature Regime | | | | 1, 2, 3, 4 |
| Altered Biogeochemical Regime | | | | 1, 3, 4 |
| Channel forming processes | | 1, 2, 3, 4 | | |
| Sediment transport dynamics | | 1, 2, 3, 4 | 1, 2, 3, 4 | |
| Sediment adsorption of contaminants and nutrients | | | | 1, 2, 3, 4 |

Tier 1 EECs are measurable characteristics that describe processes that can significantly alter the morphological or chemical characteristics within a river channel.

**Table 2. Stressors affecting Tier 2 Essential Ecosystem Characteristics (EEC) identified in the application of the conceptual model to case studies in the Canadian River (1), Colorado River (2), Columbia River (3), and Upper Mississippi and Illinois rivers (4).**

| Stressor | Tier 2 EEC (habitat) | | | | |
|---|---|---|---|---|---|
| | Spawning | Overwintering | Juvenile fish | Larval fish | Invertebrate |
| Altered riparian community | | | | | 1 |
| Channel stability | | | | | 2, 3 |
| Contaminants | 1, 3, 4 | | 4 | | |
| Discharge | | | | | 1 |
| Dissolved oxygen | 4 | 4 | 4 | | |
| Habitat fragmentation | 1, 3, 4 | | | 1, 2, 3 | 2, 3 |
| Sediment deposition | 3, 4 | 4 | 4 | 3 | 1, 2, 3 |
| Turbidity | | | | 2 | |
| Water Depth | 4 | | 4 | | |
| Water temperature | 1, 2, 3, 4 | 4 | 4 | 1, 3 | |
| Water velocity | 3, 4 | 4 | 4 | | |

Tier 2 EECs are broadly described as physical, chemical, or biological components of "habitat" that are hypothesized to have overall fitness consequences.

systems examined. However, in some rivers the altered hydrologic regime originated from other anthropogenic (e.g., water use, land use, biological community alteration) and natural (e.g., climate) drivers. Similarly, all four case studies listed an altered water temperature regime as a stressor to the biogeochemistry/thermodynamics EEC with linkages to several anthropogenic drivers (Figs 4, 6–8; S1–S4 Tables). There were also similarities across case studies with respect to the identification of interactions between Tier 1 EECs with all four case studies noting interactions between Tier 1 EEC components.

For Tier 2, there were similarities across case studies; however, the adaptation and elaboration of the components to the management goal in the case studies was apparent (Table 2). The management goal associated with the case study for the Upper Mississippi and Illinois Rivers resulted in Tier 2 EEC components (e.g., overwintering habitat) and stressors (e.g., dissolved oxygen) that were unique. Spawning habitat was identified as a Tier 2 EEC component in all the case studies and multiple stressors were identified as affecting this component in two or more of the case studies. Larval fish and invertebrate habitat were noted as Tier 2 EEC components with some similarities in stressors across case studies. Habitat fragmentation, sediment deposition, and water temperature were listed as stressors to Tier 2 EECs in all four case studies. No interactions between Tier 2 EEC components were listed for the case studies.

The adaptation of Tier 3 EECs and elaboration of the biological system related to the management goal addressed by the case studies resulted in EECs that were comprised of fish life stages ranging from eggs to adult fish, primary and invertebrate production, and biodiversity (Table 3). All Tier 3 EEC components, except biodiversity, were present in the four case studies. Not surprisingly, habitat quantity and quality were listed as stressors to all the EECs related to fish and invertebrates. Six of eight stressors or inter-tier interactions were listed as affecting fish larvae and five of eight were noted as affecting egg quantity and quality. In contrast to Tier 2, interactions were extensively noted between Tier 3 EECs and trophic level interspecific interactions were listed in all four case studies.

## Discussion

The main objective of our exploration of CMs was to impose some structure on the complex ecosystems found in large rivers and from that structure, identify gaps in monitoring

**Table 3. Stressors or inter-tier interactions affecting Tier 3 Essential Ecosystem Characteristics (EEC) identified in the application of the conceptual model to case studies in the Canadian River (1), Colorado River (2), Columbia River (3), and Upper Mississippi and Illinois rivers (4).**

| Stressor or inter-tier interaction | Tier 3 EEC | | | | | | |
|---|---|---|---|---|---|---|---|
| | Adult fish recruitment | Juvenile fish recruitment | Larval fish production | Fish egg quality/ production | Invertebrate production | Primary production | Biodiversity |
| Direct mortality | | | 1 | | | | |
| Predation/competition by invasive species | | | 1, 2, 3 | 3 | | | 4 |
| Habitat quantity/quality | 4 | 4 | 1, 2, 3 | 1, 2, 3, 4 | 1, 2, 3 | | |
| Nutrient flux | | | | | | 1, 2, 3 | |
| Trophic level interspecific interactions | 1, 2, 3, 4 | 1, 2, 3, 4 | 1, 2, 3, 4 | 1, 2, 3, 4 | 1, 2, 3, 4 | 1, 2, 3, 4 | 4 |
| Predation | | | 3 | 3 | | | |
| Mortality | 1, 3, 4 | 1, 2, 3, 4 | 1, 2, 3 | | | | |
| Fish condition | | | | 1, 3, 4 | | | |

Tier 3 EECs represent components of the hypothesized biological system upon which the cascading effects of anthropogenic and natural drivers act, and interactions occur.

information that could inform the management of fish. Comparison across our four case studies provides some insights into large rivers and the utility of the CM to identify gaps in our understanding of factors affecting fish in large rivers.

Despite large differences in the physical and ecological contexts of the river systems, the case studies demonstrated substantial commonalities in the data needed to better understand how human activities affect these systems and in the application of the CM. The general tiered structure of drivers and cascading responses through EECs worked well with the four examples. Each of the four rivers could be placed in the tiered CM to illustrate current perceptions about drivers and responses. The hierarchical CM generally increased in complexity from top to bottom. Among all rivers, there tended to be greater understanding of links from drivers to Tier 1 and Tier 2 EECs, and less understanding about linkages to Tier 3.

The strength of understanding of interactions between anthropogenic and natural drivers and EECs, and between and within EECs, varied considerably among river systems, however, resulting from both variable complexity and existing knowledge. For example, linkages from drivers to Tier 1 EECs were considered strong in the case of the Humpback Chub in the Grand Canyon, but between Tier 1 and Tier 2 only moderate. This is probably indicative of the substantial research investments in examination of physical processes in this river system [95].

Although we did not prescribe a specific approach to the CM process, the case studies employed similar strategies. Our modelling exercises started with the definition of a management goal. In all our case studies, the management goals pertained to a desired biological endpoint represented in Tier 3. After the definition of the management goal, we conceptualized interactions between drivers and EECs and between EECs with a combination of top-down and bottom-up approaches. A top-to-bottom approach to working with these models is generally consistent with a management perspective wherein anthropogenic drivers that are most directly managed in a large-river system (e.g., land and water use, etc.) cascade from top to bottom through fluxes to physical and chemical habitats, and then to biological responses. While this is generally true for anthropogenic drivers, a notable exception to the top-to-bottom management approach would be that in the U.S., there are few actions currently directed at reducing emissions affecting climate [96] which is a natural driver in our CM. Climate was

hypothesized to be a stressor in the case study application of the CM to Arkansas River Shiner management in the South Canadian River and is hypothesized to be affecting hydrologic regimes elsewhere [97, 98], but was not specifically mentioned in other case studies. The CMs can readily be modified to incorporate other factors or pathways (e.g., climate effects) as new information or perspectives become available. A bottom to top approach is equally or more valuable as it starts with the foundation of understanding about the species or community, and then seeks to identify which stressors affect population or community responses. A bottom-up approach can readily identify information gaps in linkages from ecological processes to demographic parameters [99].

The top-to-bottom and bottom-up approaches meet in the middle in Tier 2 in the concept of habitat: the resources and conditions present in an area that produce occupancy [100]. Tier 2 is critical as it has little value if it is not defined based on biological requirements or if managers lack understanding on how habitat is formed. Among our examples, the Upper Mississippi River is notable for asserting strong understanding of the linkages from land-use stressors to sediment regime to diminished overwintering habitat for native adult fishes. After that, interactions with other processes and life stages combine to increase uncertainty about whether overwintering habitat is a limiting factor in biodiversity. In contrast, the high confidence in understanding how White Sturgeon egg quality and production are linked to spawning habitat in the Columbia River Basin, provides a strong linkage upward through Tier 1 EECs and potential management actions (Fig 7). Although at times elusive, the concept of habitat is critical for linking management to biotic endpoints [101].

Large rivers are typically managed for multiple objectives, including fisheries, multi-species, or ecosystem objectives. Management decisions typically require an understanding of how management actions propagate through a river ecosystem. Although the emphasis may be on a biological endpoint (among other objectives), understanding the intermediate steps and the processes linking them, and potential interactions between processes or EEC components, can help formulate effective management strategies; especially as multiple objectives compete. In a multi-species context, the conceptual models can help identify commonalities and differences in in how stressors propagate to biota and therefore provide a basis for prioritizing monitoring efforts. In the case where species or guilds have similar habitat affinities and life histories, a dominant anthropogenic stress pathway may be hypothesized and focus on a single or few monitoring components may be justified. An example may be multiple large-river species that are known to be cued to spawn by spring flow pulses. In such a case, the characteristics of the annual hydrograph would be a dominant physical monitoring variable and biological monitoring could focus on reproductive success of one or more of the species. In the case where multiple species of concern have different reproductive strategies–for example, rheophilic species like sturgeon that may require in-channel dispersal of young to flowing habitats compared to invasive carp whose young thrive when they can disperse to lentic floodplain pools–pathways and monitoring strategies will diverge. In the latter case, it would probably not be sufficient to monitor and assess the characteristics of the annual hydrograph; instead, hydrologic metrics would need to be integrated with hydraulic and geomorphic metrics to assess where and when the different habitats would be available and could be targeted for young-of-the-year sampling. Effectively addressing multiple species would rely on detailed knowledge of life histories and how they play out on the landscape–such information is missing for many species and may need to be developed for effective design of monitoring and management actions.

Management actions intended to benefit fish in large rivers can directly or indirectly affect multiple ecosystem components. Without consideration of the effects of management on non-target ecosystem components, unintended consequences may limit management efficacy. Hypothesizing inter-tier interactions in the Tier 3 EEC (e.g., see Fig 7), can provide insight on

the potential interactions among fish species and other biological components in the context of the hierarchical CM. In all our case studies, the lumping of multiple biological interactions in Tier 3 resulted in a simplification of complex trophic interactions. For example, as Tier 3 encompasses all biological responses, it includes multiple life stages of many interacting species at varying trophic levels. Because of this, the four CMs diverged significantly at Tier 3 as components were expanded to accommodate existing understanding. Even as the Tier 3 components were expanded in complexity, they remained highly simplified views of the ecosystem. Simplification was based, in part, on the importance of key species in management goals and the experts' existing knowledge. Even though the hypothesized Tier 3 interactions in our case studies conveyed a simplification of the trophic interactions, the hypothesized interactions do suggest the need for information that clarifies the trophic interactions and effects of Tier 3 EEC components on the biological endpoint. If desired or warranted, the Tier 3 EEC could be elaborated to capture more complexity. For example, in Fig 7, the Tier 3 inter-tier interaction between anadromous and resident fishes and white sturgeon larvae could expanded to include interactions with specific fish species. Monitoring can help clarify the effects of management actions, including on non-target ecosystem components, but only if data are collected to characterize key ecosystem processes that could affect the outcome. The process of considering and elucidating Tier 3 EEC interactions can help identify the non-target ecosystem components that could be affected when managing for a specific biological endpoint.

The CMs explored here also provide a framework for considering return on science investments. The knowledge needed for effective management of large rivers can be gained by monitoring intermediate endpoints along the cascade, but the type of information and costs vary widely. Costs for monitoring Tier 1 EECs can be high but some programs are already in place. For example, large rivers are likely to have monitoring infrastructure installed for Tier 1 monitoring of discharge and temperature regimes, with varying potential for monitoring sediment transport and water quality. Investment at Tier 2 may emphasize physical processes and habitats that can be measured at relatively low cost, assuming that habitats are adequately defined based on biological criteria. In larger rivers, Tier 2 habitat assessments can be more cost effective compared to smaller rivers because they can rely on automated data collection through hydroacoustics and remote sensing [101]. As discussed above, habitat assessments have value only to the extent that they are based on well-defined biological requirements; it is notable that some large-river management efforts have found that relatively simple habitat models are useful to predict biological responses [102]. At Tier 3, costs can increase substantially because of structural uncertainties (i.e., which life stages, which species are most important to monitor) and because of the inherent uncertainties of monitoring fish in large river systems where detection probability can be low and highly variable [36, 103]. Generally, the cost of monitoring increases from Tier 1 to Tier 3 in the CM hierarchy; at the same time, the relevance of information to decision making is typically greater for biological responses depicted in Tier 3 [104].

Because both costs and information benefits increase from Tier 1 to Tier 3 in the CM hierarchy, it is difficult to generalize about where the benefit:cost ratio would be optimized. Indeed, as discussed by Jacobson and Berkley [30], the decision about where in the hierarchy monitoring resources would get the highest return on investment may depend more directly on managers' and stakeholders' perceptions about risks of acting with incomplete information. For example, the details of how a fish's reproductive strategy depends on the nuances of a seasonal hydrograph may not be known, but stakeholders may believe strongly that the natural hydrograph was functional for the species and therefore monitoring of the flow regime will have the highest return on investment and, by extension, restoration of the flow regime is likely to have the most positive effects. On the other hand, in systems where stakeholders

opinions are divided or socio-economic values would be compromised by a return to a natural flow regime, managers may be required to demonstrate more precisely how elements of the flow regime propagate to species' benefits [105]. Thus, once information needs are identified and the availability of data is assessed, the costs of collecting the information should be placed in a socioeconomic context (e.g., see [30]).

The development of the CMs described in this manuscript can be a first step in application of structured decision-making (SDM) and its iterative form-adaptive management (AM) processes [106, 107]. Structured decision-making is a stakeholder driven process by which a problem can be defined with conceptual models and decomposed into decision components that include the problem context, stakeholder objectives, potential management actions, consequences of those actions on the objectives, and trade-offs related to different decisions (actions) [107–109]. One primary focus of SDM is the identification of uncertainties such as those identified in the CMs for the case studies in this paper [110]. Quantification of the influence of decision relevant uncertainties can be modeled using sensitivity analysis and other techniques and ranked [107, 109, 111]. In addition, the quantitative techniques available to assist in solving complex ecological problems are robust and range in complexity from consequences tables to Bayesian models to dynamic optimization models [107–109, 112, 113]. The SDM process is often used as the set-up phase for adaptive management which includes monitoring over time to reduce uncertainty related to how management will influence important outcomes (e.g. fish population status; [109, 112].

The CM may also help to identify which processes or components are amenable to a field monitoring effort and which are more aptly addressed through laboratory or mesocosm experiments. For example, if it is hypothesized that the condition of age 1+ Arkansas Shiners is a critical determining factor in egg quality or production (Fig 4), it could be determined that the best approach to developing a quantitative relation between condition and eggs is through a controlled laboratory experiment rather than field-based monitoring. The CM helps to visualize where different types of information may be applied within a decision-making framework.

A large-river CM may also serve as a precursor to computational ecological or population models [30]. Similar questions about how monitoring and other science efforts should be distributed among EECs and processes can be addressed iteratively by carrying out sensitivity analyses in a modeling framework. Indeed, given substantial uncertainties associated with monitoring data, computation modeling can be considered a necessary component of large-river monitoring and evaluation systems [99, 114].

## Conclusions

We found the process of conceptualizing the relationships between and within EECs fostered a critical assessment of what we know about factors affecting the management endpoint. By visualizing how EEC drivers directly and indirectly affect management endpoints, our CM identified critical information gaps and uncertainties that, if resolved, could improve our understanding of how to best meet management objectives. The process of conceptualizing the EEC relationships affecting fish in large rivers could help to structure, or restructure, monitoring programs around scientifically sound monitoring questions, promote the selection of relevant ecological indicators that characterize resource condition or management outcomes, and facilitate communication and information sharing within and between organizations managing or researching management endpoints. Ultimately, understanding the mechanisms by which EECs influence large-river fishes will improve the effectiveness of restoration and management actions.

As shown with our case studies, our CM is flexible and applicable to a wide range of river systems with different anthropogenic drivers and management goals. We feel our CM provides a generic structure that scientists can adapt to their unique needs. By not being overly prescriptive, for example, with respect to the components of the Tier 2 and 3 EEC components, scientists can adapt the CM to different biological communities and management endpoints. By doing so, we feel that users have the flexibility to place their management questions in the context of EECs that are specific to their large-river system. We provide a stepwise procedure to facilitate the application of our conceptual model to other river systems and management goals (Table 4).

**Table 4. The recommended steps to conduct the conceptual modelling exercise.**

| Step | Component |
|---|---|
| | **Initial problem definition and approach** |
| 1 | Identify fundamental goal (management endpoint) that is the focus of monitoring or that monitoring information could inform. This is likely done in collaboration with agencies and stakeholders. |
| 2 | Convene expert practitioners with range of expertise in aquatic ecology, fisheries, and physical river processes. |
| | **Assemble critical biological information** |
| 3 | Identify what is known about life history needs or known bottlenecks for the biological components of management concern. Detailed understanding of the biology underlies the structure and assumptions of the process. |
| | **Conceptual Model Tier 1: Understand natural and altered system fluxes** |
| 4 | Identify and prioritize natural and anthropogenic drivers that result in natural system fluxes that affect Tier 1 EECs in your river system. |
| 5 | Hypothesize the nature of natural system fluxes arising from natural and anthropogenic drivers that are affecting Tier 1 EECs (e.g., altered hydrologic regime). |
| 6 | Identify intra-tier interactions for Tier 1 EECs given the natural system fluxes acting on Tier 1. |
| 7 | Hypothesize the strength of understanding of how the natural and altered system fluxes are affecting Tier 1 EECs and simplify the conceptual model to those components that are uncertain and/or important. |
| | **Conceptual Model Tier 2: Understand natural and altered habitat characteristics** |
| 8 | Elaborate Tier 2 EECs to reflect knowledge of important habitat characteristics given expert knowledge of life history needs or bottlenecks identified. |
| 9 | Hypothesize how the effects of natural system fluxes affecting Tier 1 EECs are transmitted to Tier 2 EECs. |
| 10 | Identify intra-tier interactions for Tier 2 EECs. |
| 11 | Hypothesize the strength of understanding of how the Tier 1 EECs that are being affected by natural system fluxes are affecting Tier 2 EECs and simplify the conceptual model to those components that are uncertain and/or important. |
| | **Conceptual Model Tier 3: Understand propagation of stresses to biological components** |
| 12 | Elaborate Tier 3 EECs to include biological system components that affect the management endpoint. |
| 13 | Hypothesize how stresses on Tier 2 EECs arising from Tier 1 EECs are transferred to the biological systems depicted in Tier 3 EECs. |
| 14 | Identify intra-tier interactions for Tier 3 EECs. |
| 15 | Hypothesize the strength of understanding of how stresses on Tier 2 EECs arising from Tier 1 EECs are affecting the biological systems depicted in Tier 3 EECs and simplify the conceptual model to those components that are uncertain and/or important. |
| | **Process check** |
| 16 | Seek peer review of conceptual model |
| | **Information assessment** |
| 17 | For each inter- and intra-tier interaction identified, use expert opinion to hypothesize the information needed to describe the interaction and whether the data required to understand the information needs are available, insufficient, or not available. |
| | **Monitoring design** |

*(Continued)*

**Table 4.** (Continued)

| Step | Component |
|------|-----------|
| 18 | For information needs that were classified as insufficient or not available, characterize the spatial and temporal scales at which data should be collected to make inferences that support the evaluation of the management goal or objective. |

The exact sequencing of the steps is flexible. For example, some may prefer a bottom-up approach while others may prefer to work from the top-down.

Although the case studies addressed management issues that were river or basin specific, there were similarities relative to information needs and data availability. For example, in most systems information on river discharge and water temperature were needed and available. Conversely, information regarding trophic relationships and the habitat requirements of larval fishes were generally lacking. This result suggests that there is a need to better understand a set of common factors across large-river systems.

## Supporting information

**S1 Appendix.**
(DOCX)

**S1 Fig. Daily discharge ($m^3$/s) patterns from 1938 to 2019 in the Canadian River, Oklahoma near Canadian, TX (data are available at https://waterdata.usgs.gov/nwis/uv/?site_no=07228000).**
(DOCX)

**S2 Fig. Hydrograph of the Colorado River at Lees Ferry, AZ from 1930 to 2019 showing pre- and post-Glen Canyon Dam closure in 1964 (dashed line) mean monthly discharge ($m^3$/s), which transitions from seasonally stochastic to a more homogeneous regime focusing on anthropogenic interests (data are available at https://waterdata.usgs.gov/nwis/dv/?site_no=09380000).**
(DOCX)

**S3 Fig. The spatial and temporal scales of the management goal, the scientific inferences needed to inform the management goal, and that data collection needs to occur to support the inferences for monitoring information needs identified as requiring additional data in the case study addressing Humpback Chub recruitment in the Colorado River between Glen Canyon Dam and Lake Mead, Arizona (see S2 Table for additional detail).** A:Tier 2 EEC = larval Humpback Chub habitat; Stressors = habitat fragmentation, turbidity and Tier 2 EEC = Humpback Chub spawning habitat; Stressor = water temperature and Tier 3 EEC = insect production; Stressor = benthic macroinvertebrate habitat quantity and quality and Tier 3 EECs = all; Inter-tier interaction = trophic level interactions; B:Tier 3 EEC = larval Humpback Chub production; Stressor = larval Humpback chub habitat quantity and quality; C: Tier 3 EEC = Humpback Chub egg quality and production; Stressors = Humpback Chub spawning habitat quantity and quality; D:Larval Humpback Chub production; Inter-tier interaction = mortality of Humpback Chub eggs; E:Tier 3 EEC = larval Humpback Chub production; Stressor = predation by invasive species and Tier 3 EEC = Primary production; Stressor = nutrient flux; F:Tier 1 EEC = biogeochemistry/thermodynamics; Inter-tier interaction = sediment adsorption of contaminants and nutrients; G:Tier 3 EEC = Humpback

Chub age-0 recruitment; Inter-tier interaction = mortality of larval Humpback Chub.
(DOCX)

**S4 Fig. Proportion of total annual Columbia River discharge at The Dalles, OR occurring in the month of June from 1879 to 2015 (data available at https://waterdata.usgs.gov/nwis/dv/?site_no=14105700).**
(DOCX)

**S5 Fig. The spatial and temporal scales of the management goal, the scientific inferences needed to inform the management goal, and that data collection needs to occur to support the inferences for monitoring information needs identified as requiring additional data in the case study addressing White Sturgeon recruitment in the Columbia River (see S3 Table for additional detail).** A:Tier 1 EEC = biogeochemistry/thermodynamic; Stressor = altered biogeochemical regime and Tier 2 EEC = benthic macroinvertebrate habitat; Stressors = channel stability, sediment deposition, fragmentation and Tier 2 EEC = Larval White Sturgeon habitat; Stressors = habitat fragmentation sediment deposition, water temperature and Tier 2 EEC = White Sturgeon spawning habitat; Stressors = contaminants, sediment deposition and Tier 3 EEC = White Sturgeon egg quality and production; Stressor = predation by invasive species and Tier 3 EEC = White Sturgeon larvae production; Stressor = predation by invasive species and Tier 3 EEC = benthic macroinvertebrate production; Stressor = benthic macroinvertebrate habitat quantity and quality; B:Tier 1 EEC = sediment transport; Stressors = altered sediment regime, altered hydraulic regime and Tier 1 EECs = channel morphology/hydraulics, sediment transport; Stressor = altered hydraulic regime and Tier 3 EEC = primary production; Stressor = nutrient fluxes; C:Tier 1 EEC = channel morphology/hydraulics, sediment transport; Stressor = altered hydraulic regime and Tier 1 EEC = channel morphology/hydraulics, sediment transport; Inter-tier interaction = sediment transport dynamics; D:Tier 1 EEC = channel morphology/hydraulics, sediment transport; Stressor = altered hydraulic regime and Tier 1 EEC = channel morphology/hydraulics, sediment transport; Inter-tier interaction = sediment transport dynamics; E:Tier 1 EEC = biogeochemistry/thermodynamics; Inter-tier interaction = sediment adsorption of contaminants and nutrients; F: Tier 3 EEC = White Sturgeon larvae production; Stressors = larval White Sturgeon habitat quantity and quality and Tier 3 EEC = White Sturgeon larvae production, age-0 White Sturgeon recruitment; Inter-tier interaction = mortality and White Sturgeon egg quality and production; Inter-tier interaction = predation of White Sturgeon eggs by native fish and Tier 3 EECs = White Sturgeon larvae production; Inter-tier interaction = predation of White Sturgeon larvae by native fish and Tier 3 EECs = all; Inter-tier interactions = trophic level interactions.
(DOCX)

**S6 Fig. Largemouth bass (*Micropterus salmoides*) data from the Pool 13 of the Upper Mississippi River (A, B) and the La Grange Pool of the Illinois River (C, D).** The two river reaches are roughly the same latitude, but the La Grange Pool is more limited in overwintering habitat. Population abundance is presented in panels A and C where each point is an individual fish cumulatively caught with standardized day time electrofishing annually from 1993 to 2017 (data available at https://umesc.usgs.gov/data_library/fisheries/fish1_query.shtml). The dashed triangle highlights 'missing' >400 mm size classes since 2000 in the La Grange Pool. Population size structure is indexed by proportional stock density (PSD) is presented in panels B and D with the dashed line showing trends in the largest size classes from 1993 to 2017 (data available at https://umesc.usgs.gov/data_library/fisheries/graphical/fish_front.html).
(DOCX)

**S7 Fig. The spatial and temporal scales of the management goal, the scientific inferences needed to inform the management goal, and that data collection needs to occur to support the inferences for monitoring information needs identified as requiring additional data in the case study addressing native fish biodiversity and habitat diversity in the Mississippi and Illinois rivers (see S4 Table for additional detail).** A:Tier 1 EEC = sediment transport; Stressor = altered hydraulic regime and Tier 1 EEC = biogeochemistry/thermodynamics; Stressor = altered biogeochemical regime and Tier 1 EEC = biogeochemistry/thermodynamics; Inter-tier interaction = sediment adsorption of contaminants and nutrients; B: Tier 2 EEC = adult native fish overwintering habitat; Stressors = water velocity, water temperature, dissolved oxygen, sediment deposition and Tier 2 EEC = juvenile native fish habitat; Stressors = water depth, water velocity, water temperature, dissolved oxygen, contaminants, sediment deposition and Tier 2 EEC = native fish spawning habitat; Stressors = water depth, water velocity, habitat fragmentation, sediment deposition, water temperature, dissolved oxygen, contaminants and Tier 3 EEC = adult and juvenile native fish recruitment; Inter-tier interaction = mortality and Tier 3 EEC = all; Stressors = invasive species and Tier 3 EEC = all; Inter-tier interaction = trophic level interactions; C:Tier 1 EEC = channel morphology/hydraulics; Inter-tier interaction = channel forming processes; D:Tier 1 EEC = channel morphology/hydraulics, sediment transport; Inter-tier interaction = sediment transport dynamics; E:Tier 3 EEC = adult native fish recruitment; Stressor = adult native fish overwintering habitat quantity and quality and Tier 3 EEC = juvenile native fish recruitment; Stressor = juvenile native fish habitat quantity and quality and Tier 3 EEC = native fish egg quality and production; Stressor = spawning habitat quantity and quality.
(DOCX)

**S1 Table. Summary of information needs identified in the Conceptual Model describing factors affecting the recruitment of the Arkansas River Shiner in the South Canadian River, OK (Fig 4; this publication) by Essential Ecosystem Characteristic (EEC) Tier, EEC, and stressor or within Tier interactions and an assessment of the status of existing information that could be used to address the information needs.**
(DOCX)

**S2 Table. Summary of information needs identified in the conceptual model describing factors affecting the recruitment of the Humpback Chub in the Colorado River, Arizona (Fig 6; this publication) by Essential Ecosystem Characteristic (EEC) Tier, EEC, and stressor or inter-tier interactions and an assessment of the status of existing information that could be used to address the information needs.**
(DOCX)

**S3 Table. Summary of information needs identified in the conceptual model describing factors affecting the recruitment of age-0 White Sturgeon in the Columbia River (Fig 7; this publication), by Essential Ecosystem Characteristic (EEC) Tier, EEC, and stressor or inter-tier interactions and an assessment of the status of existing information that could be used to address the information needs.**
(DOCX)

**S4 Table. Summary of information needs identified in the conceptual model describing factors affecting the restoration and maintenance of native fish biodiversity and habitat quantity and quality in the Upper Mississippi and Illinois rivers (Fig 8; this publication), by Essential Ecosystem Characteristic (EEC) Tier, EEC, and stressor or inter-tier interactions and an assessment of the status of existing information that could be used to address**

**the information needs.**
(DOCX)

## Acknowledgments

We thank Megan Dethloff of the Pacific Northwest Aquatic Monitoring Partnership for her role in organizing the logistics associated with the workshop and conference calls needed to develop this manuscript. David Ward from the U.S. Geological Survey's Grand Canyon Monitoring and Research Center provided many valuable comments and suggestions that improved the manuscript. Any use of trade, firm, or product names is for descriptive purposes only and does not imply endorsement by the U.S. Government. An animal care and use protocol was not required for this research.

## Author Contributions

**Conceptualization:** Timothy D. Counihan, Kristen L. Bouska, Shannon K. Brewer, Robert B. Jacobson, Andrew F. Casper, Colin G. Chapman, Ian R. Waite, Kenneth R. Sheehan, Mark Pyron, Elise R. Irwin, Karen Riva-Murray, Alexa J. McKerrow, Jennifer M. Bayer.

**Formal analysis:** Timothy D. Counihan, Kristen L. Bouska, Shannon K. Brewer, Robert B. Jacobson.

**Funding acquisition:** Timothy D. Counihan, Alexa J. McKerrow, Jennifer M. Bayer.

**Investigation:** Timothy D. Counihan, Kristen L. Bouska, Shannon K. Brewer, Robert B. Jacobson.

**Methodology:** Timothy D. Counihan, Kristen L. Bouska, Shannon K. Brewer, Robert B. Jacobson, Andrew F. Casper.

**Project administration:** Timothy D. Counihan, Jennifer M. Bayer.

**Supervision:** Jennifer M. Bayer.

**Visualization:** Timothy D. Counihan, Kristen L. Bouska, Shannon K. Brewer, Robert B. Jacobson, Andrew F. Casper, Colin G. Chapman, Ian R. Waite, Kenneth R. Sheehan.

**Writing – original draft:** Timothy D. Counihan, Kristen L. Bouska, Shannon K. Brewer, Robert B. Jacobson, Andrew F. Casper, Colin G. Chapman, Kenneth R. Sheehan, Jennifer M. Bayer.

**Writing – review & editing:** Timothy D. Counihan, Kristen L. Bouska, Shannon K. Brewer, Robert B. Jacobson, Andrew F. Casper, Colin G. Chapman, Ian R. Waite, Mark Pyron, Elise R. Irwin, Karen Riva-Murray, Alexa J. McKerrow, Jennifer M. Bayer.

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
