## [Decision Letter · Decision Letter 0]

16 Jun 2021

PONE-D-21-12444

Identifying monitoring information needs that support the management of fish in large rivers

PLOS ONE

Dear Dr. Counihan,

Thank you for submitting your manuscript to PLOS ONE. After careful consideration, we feel that it has merit but does not fully meet PLOS ONE’s publication criteria as it currently stands. Therefore, we invite you to submit a revised version of the manuscript that addresses the points raised during the review process.

We have now received reviews from two anonymous.

As you can see from their relatively brief comments, they both agree on the value of the proposed conceptual model for identifying information gaps and guide monitoring in conservation science. However, they also point out some important shortcomings that need your attention. The same set of issues were raised by both reviewers.

Reviewer 1 wonders about the utility of such conceptual models in the more realistic case of multiple species contexts. How could the model be expanded in this regard, besides the case of simple richness measures? In addition, how could the model help in actually prioritising or ranking the variables or interactions identified? Finally, reviewer 1 also noticed that an important stressor related to fragmentation and connectivity is only marginally discussed and included in the model. This is critical for meta-population dynamics and should be given more emphasis.

Reviewer 2, similarly raises the critical issue of the multi-species context, and how the needs of different species could be simultaneously identified. Therefore, I advice to expand the Discussion in this regard, eventually acknowledging limitations and suggesting future research needs. Reviewer 2 also wonders how such models could effectively guide restoration and decision-making beyond monitoring needs; if the relative importance of data gaps and interactions is not quantified (e.g. via a cost-benefit analysis), how could it help prioritise the focus of monitoring and action?

Besides carefully responding to each reviewers' comments and modify the manuscript accordingly, I also suggest to simplify the manuscript, which feels rather long. Perhaps some of the background information from each case study could be included as supplementary or shortened. Also, caption from Fig.5 (and sister-figures) is rather hard to digest for the reader. I wonder if this could be simplified as well.

I believe that the issues raised by the reviewers are valid and should considered with great care before we can accept the manuscript, even though both proposed 'minor revisions' are needed.

We look forward to your revision in due time.

We look forward to receiving your revised manuscript.

Kind regards,

Stefano Larsen

Academic Editor

PLOS ONE

Journal Requirements:

Additional Editor Comments (if provided):

Reviewers' comments:

Reviewer's Responses to Questions

**Comments to the Author**

1. Is the manuscript technically sound, and do the data support the conclusions?

Reviewer #1: Yes

Reviewer #2: Partly

2. Has the statistical analysis been performed appropriately and rigorously? 

Reviewer #1: N/A

Reviewer #2: N/A

3. Have the authors made all data underlying the findings in their manuscript fully available?

Reviewer #1: Yes

Reviewer #2: Yes

4. Is the manuscript presented in an intelligible fashion and written in standard English?

Reviewer #1: Yes

Reviewer #2: Yes

5. Review Comments to the Author

Reviewer #1: In this study a conceptual model is used to aid the development of best practices of large river monitoring programs. The model was developed based on former scientific works and during scientists workshop negotiations. Case study applications prove that the application of this complex conceptual model can be useful to identify critical information gaps, which can then be used to develop management and monitoring objectives.

I like the approach of developing such conceptual models, which can reveal information gaps, and think that the model in general can be useful to adopt across large river systems with some refinements and local adaptations. Consequently, I believe showing such an approach can provide useful information for the readers.

What I lack is to show more convincingly how such complex models can be used for multispecies systems, where not only the requirements of a single species is evaluated, which in fact the more realistic situation. How can individual species level models be put together to provide meaningful information for management? It would be useful to discuss this in more detail in the Discussion section. Also a critical issue which should be briefly discussed is how the identified critical target variables should be prioritized, especially in a multispecies systems, where several variables will appear. Development of this section could convince the reader and could clearly show the applicability of such conceptual models by management.

Although channel morphology/hydraulics may contain fragmentation/connectivity issues this should be made more clear in the material, because this is one of the most critical issue, which determine fish (meta)population or metacommunity dynamics. In fact fragmentation is often used as one of the most critical variable of anthropogenic drivers and as such is a critically important target to mitigate by management. However, it does not appear either in Fig 2 or Fig 3, but only on the case study figures belonging to morphology/hydraulics TIER1 components.

Reviewer #2: Dear Editor,

This study demonstrates how a conceptual model can be used to identifying knowledge gaps in the mechanisms by which Essential Ecosystem Characteristics influence large-river fish species in the USA. These gaps should then be filled to improve the effectiveness of restoration and management.

I agree on the value of these conceptual models to identify knowledge gaps and inform decisions on what to monitor to fill them and allow a better understanding of the system and, therefore, enhance our capacity to manage them adequately. However, I disagree with some of the arguments:

- The conceptual model represents potential interactions across different structural element of the river system but does not allow quantitative evaluations of strength of those interaction. As such, the value of is conceptual model is limited to identifying knowledge gaps and cannot be used to evaluate the relative importance of each interaction. Therefore, this conceptual model should only be used for identifying knowledge gaps and not for decision-making, as argued (see L771-773), beyond monitoring.

- The conceptual model lacks a cost analysis to evaluate the most efficient way of filling knowledge gaps. Some of the gaps might be more difficult/ costly or even feasible to fill. Without such analysis we can only identify the gaps but cannot prioritise where to focus monitoring on a cost-effective way and just confirm where gaps exist.

- Three of the case studies present conceptual models for individual species. While I see the value of developing these conceptual models for charismatic endangered species, I wonder how feasible/ useful it would be this method when facing management needs for many species simultaneously. One of the case studies does present a conceptual model for the full fish community, but focused on diversity, rather than individual species, so no information of particular species issues are addressed. Would it be feasible to elaborate a conceptual model that addressed all individual species needs/ issues simultaneously? This would allow identifying knowledge gaps common to multiple species simultaneously.

Minor comments:

- L304 & 369. What does ATV stand for?

- The manuscript is quite long, especially because of the description of each case study. It would be good to present the information of these case studies in a more synthetic way (maybe on a table?).

6. PLOS authors have the option to publish the peer review history of their article (what does this mean?). If published, this will include your full peer review and any attached files.

Reviewer #1: No

Reviewer #2: No

---

## [Author Response · Author response to Decision Letter 0]

25 Aug 2021

PONE-D-21-12444

Identifying monitoring information needs that support the management of fish in large rivers

PLOS ONE

Reviewer 1 wonders about the utility of such conceptual models in the more realistic case of multiple species contexts. How could the model be expanded in this regard, besides the case of simple richness measures? In addition, how could the model help in actually prioritising or ranking the variables or interactions identified? Finally, reviewer 1 also noticed that an important stressor related to fragmentation and connectivity is only marginally discussed and included in the model. This is critical for meta-population dynamics and should be given more emphasis.

Reviewer 2, similarly raises the critical issue of the multi-species context, and how the needs of different species could be simultaneously identified. Therefore, I advice to expand the Discussion in this regard, eventually acknowledging limitations and suggesting future research needs. Reviewer 2 also wonders how such models could effectively guide restoration and decision-making beyond monitoring needs; if the relative importance of data gaps and interactions is not quantified (e.g. via a cost-benefit analysis), how could it help prioritise the focus of monitoring and action?

Besides carefully responding to each reviewers' comments and modify the manuscript accordingly, I also suggest to simplify the manuscript, which feels rather long. Perhaps some of the background information from each case study could be included as supplementary or shortened. Also, caption from Fig.5 (and sister-figures) is rather hard to digest for the reader. I wonder if this could be simplified as well.

Response: Thank you for the opportunity to revise our manuscript. We have tried to address comments of Reviewer 1 and 2 below and in the revised manuscript. To shorten and simplify the manuscript, we relocated some of the contextual information from the case studies and moved it to an Appendix in Supplemental Information. With respect to Fig 5 and sister figures, we have discussed trying to simplify the figure captions but have not come up with a good solution. There is a lot of information contained within the figures and feel that further generalizations would not be clarifying. We do, however, acknowledge that the figure caption format is awkwardly long. What we propose is that we retain Fig 5 in the main body of the text as an example, and then move subsequent sister figures to Supplemental Materials. Please let us know if this satisfies your and the reviewer’s comments to reduce the length of the manuscript.

Comments to the Author

5. Review Comments to the Author

Reviewer #1: In this study a conceptual model is used to aid the development of best practices of large river monitoring programs. The model was developed based on former scientific works and during scientist’s workshop negotiations. Case study applications prove that the application of this complex conceptual model can be useful to identify critical information gaps, which can then be used to develop management and monitoring objectives.

I like the approach of developing such conceptual models, which can reveal information gaps, and think that the model in general can be useful to adopt across large river systems with some refinements and local adaptations. Consequently, I believe showing such an approach can provide useful information for the readers.

1) What I lack is to show more convincingly how such complex models can be used for multispecies systems, where not only the requirements of a single species is evaluated, which in fact the more realistic situation. How can individual species level models be put together to provide meaningful information for management? It would be useful to discuss this in more detail in the Discussion section.

Response: Thank you for this comment. We have added text in the discussion that describes how the CM could contribute to our understanding of the need for a multi-species context in management activities. Please see: L645-691

2) Also a critical issue which should be briefly discussed is how the identified critical target variables should be prioritized, especially in a multispecies systems, where several variables will appear. Development of this section could convince the reader and could clearly show the applicability of such conceptual models by management.

Response: Thank you for this insight. We have provided language in the discussion that describes how the CM could be used to identify critical target variables in a multispecies context. Please see: L650-668

3) Although channel morphology/hydraulics may contain fragmentation/connectivity issues this should be made more clear in the material, because this is one of the most critical issue, which determine fish (meta)population or metacommunity dynamics. In fact fragmentation is often used as one of the most critical variable of anthropogenic drivers and as such is a critically important target to mitigate by management. However, it does not appear either in Fig 2 or Fig 3, but only on the case study figures belonging to morphology/hydraulics TIER1 components.

Response: Thank you. We agree with your assessment. We have further emphasized the importance of fragmentation by specifically mentioning it in the manuscript section describing the CM form. Please see: L223-234. Also, in Table 2, habitat fragmentation is emphasized as affecting multiple facets of the CM in multiple river systems and is mentioned in the text describing Table 2. This result indicates the need to better study the effects of habitat fragmentation on multiple biotic components.

Reviewer #2: Dear Editor,

This study demonstrates how a conceptual model can be used to identifying knowledge gaps in the mechanisms by which Essential Ecosystem Characteristics influence large-river fish species in the USA. These gaps should then be filled to improve the effectiveness of restoration and management.

I agree on the value of these conceptual models to identify knowledge gaps and inform decisions on what to monitor to fill them and allow a better understanding of the system and, therefore, enhance our capacity to manage them adequately. However, I disagree with some of the arguments:

1) The conceptual model represents potential interactions across different structural element of the river system but does not allow quantitative evaluations of strength of those interaction. As such, the value of is conceptual model is limited to identifying knowledge gaps and cannot be used to evaluate the relative importance of each interaction. Therefore, this conceptual model should only be used for identifying knowledge gaps and not for decision-making, as argued (see L771-773), beyond monitoring.

Response: This is an excellent point and we have removed the statement in L771-773 and elaborated on the considerations that need to be accounted for, and the difficulties with, assessing benefit:cost ratios. Please see: L692-726. Also, you are correct that the CM does not provide quantitative evaluations of the strength of the relations. We do acknowledge this and suggest that the CM could provide a basis for developing quantitative assessments in L749-754 and have added language that describes how the CMs could be the basis for Structural Decision Making and Adaptive Management processes (see L727-741).

2) The conceptual model lacks a cost analysis to evaluate the most efficient way of filling knowledge gaps. Some of the gaps might be more difficult/ costly or even feasible to fill. Without such analysis we can only identify the gaps but cannot prioritise where to focus monitoring on a cost-effective way and just confirm where gaps exist.

Response: Thank you for this comment. In addition to addressing the benefit:cost issue above, we have provided language in the discussion that describes how the CM could be used to identify critical target variables in a multispecies context but that there are critical uncertainties that need to be considered. See: L650-668. We agree with you about prioritizing based on cost effectiveness but respectfully suggest that the CM could provide information that would suggest where to focus monitoring effort.

3) Three of the case studies present conceptual models for individual species. While I see the value of developing these conceptual models for charismatic endangered species, I wonder how feasible/ useful it would be this method when facing management needs for many species simultaneously. One of the case studies does present a conceptual model for the full fish community, but focused on diversity, rather than individual species, so no information of particular species issues are addressed. Would it be feasible to elaborate a conceptual model that addressed all individual species needs/ issues simultaneously? This would allow identifying knowledge gaps common to multiple species simultaneously.

Response: Thank you for this comment. We have added text in the discussion that describes how the CM could contribute to our understanding of the need for a multi-species context in management activities. Please see: Please see: L645-691

4) Minor comments:

- L304 & 369. What does ATV stand for?

Response: Thank you for bringing this to our attention. ATV stands for all-terrain vehicle. We have removed the acronym from the revision.

- The manuscript is quite long, especially because of the description of each case study. It would be good to present the information of these case studies in a more synthetic way (maybe on a table?).

Response: Thank you for the comment. Per your and the Associate Editor’s recommendation we have pulled out some of the contextual information from the case studies and moved the information to an Appendix in Supplemental Information. We have also moved three figures and associated captions to the supplemental information section.

---

## [Decision Letter · Decision Letter 1]

10 Nov 2021

PONE-D-21-12444R1

Identifying monitoring information needs that support the management of fish in large rivers

PLOS ONE

Dear Dr. Counihan,

Thank you for submitting your manuscript to PLOS ONE. After careful consideration, we feel that it has merit but does not fully meet PLOS ONE’s publication criteria as it currently stands. Therefore, we invite you to submit a revised version of the manuscript that addresses the points raised during the review process.

Dear Dr Counihan,

first of all, I apologize for the very long processing time of you manuscript. We have now obtained comments from two additional reviewers; they were aware that the manuscript was already revised and they also had access to the original (first round) comments. 

Both reviewers clearly see the value of this work. Reviewer 3 makes an important point regarding the usability of your approach beyond the specific case studies and the scientists involved. In order to facilitate the uptake of your approach to a wider audience, perhaps a step-by-step guide or a concise follow-through instruction box could really help. Reviewer 4 has only minor comments and suggests edits to aid readability of the manuscript. Please note that Reviewer 4 has included an attachment. 

We will be pleased to accept this paper once you considered both reviewers comments carefully.

Looking forward to receive the last revision.

Many thanks for your patience

We look forward to receiving your revised manuscript.

Kind regards,

Stefano Larsen

Academic Editor

PLOS ONE

Journal Requirements:

Reviewers' comments:

Reviewer's Responses to Questions

**Comments to the Author**

1. If the authors have adequately addressed your comments raised in a previous round of review and you feel that this manuscript is now acceptable for publication, you may indicate that here to bypass the “Comments to the Author” section, enter your conflict of interest statement in the “Confidential to Editor” section, and submit your "Accept" recommendation.

Reviewer #3: (No Response)

Reviewer #4: (No Response)

2. Is the manuscript technically sound, and do the data support the conclusions?

Reviewer #3: Yes

Reviewer #4: Yes

3. Has the statistical analysis been performed appropriately and rigorously? 

Reviewer #3: N/A

Reviewer #4: Yes

4. Have the authors made all data underlying the findings in their manuscript fully available?

Reviewer #3: (No Response)

Reviewer #4: Yes

5. Is the manuscript presented in an intelligible fashion and written in standard English?

Reviewer #3: Yes

Reviewer #4: Yes

6. Review Comments to the Author

Reviewer #3: This revised manuscript defines and explores a conceptual model setting a structured framework that aims to identify gaps in monitoring information that could inform the management of fish in large rivers. After reading the manuscript, I still have the general feeling that something is lacking to make a more useful contribution to river and fish managers. As it stands, I feel that the manuscript is the result of trying to synthetize discussions and outcomes from the held workshops. But the manuscript tries to offer more than that. If I understand well, the authors want to provide a model to be followed so that, at the end, the model user comes out with the variable(s) that would need to be measured in order to fill the knowledge gap(s) that we need to improve the management of a river or a species. If the authors want their model to be used by other researchers and managers than those that participated to the workshops and those that work on the four study cases, a clear guidance through the different steps should be provided, just as a manual. Otherwise, the manuscript is just the result of a brainstorming that the authors want to share in a paper. I am NOT saying that the information provided in the manuscript is useless or has no rigor or significance. On the contrary, I think the manuscript brings interesting evidence and synthesis. I am just saying that if the authors want their model to be used, they need to provide clear step-by-step guidance. I think this is more useful than case studies if they want the model to be largely used.

PlosOne is publishing papers on the basis of methodological rigor and high ethical standards. Based on that and since the manuscript is rather conceptual, I think the manuscript can be published in this journal without further changes, so that my above general comment is only a suggestion.

L. 114-116. Something is missing in this sentence?

L.264. add “hierarchically” after “the spatial scales considered are”

L.512-515. This is repeated in many figures. Consider reducing this to avoid repetitions

Resolution of the figures must be improved

Reviewer #4: In this manuscript the authors used a conceptual model to help management of large river monitoring programs in identifying knowledge gaps.

I believe the authors have addressed very well the comments from the first round of review. For this reason the manuscript story and approach sound solid with a good description for the four cases, highlighting the main potential factors of disturbance and knowledge gaps in large rivers as well as when informations are not enough and more data are necessary.

I have included in the attached pdf some small comments to help the authors to improve the readability, and I believe that the authors would not need any additional comment on the main manuscript concept.

Best regards

7. PLOS authors have the option to publish the peer review history of their article (what does this mean?). If published, this will include your full peer review and any attached files.

Reviewer #3: No

Reviewer #4: No

---

## [Author Response · Author response to Decision Letter 1]

23 Dec 2021

PONE-D-21-12444R1

Identifying monitoring information needs that support the management of fish in large rivers

PLOS ONE

Dear Dr. Counihan,

Thank you for submitting your manuscript to PLOS ONE. After careful consideration, we feel that it has merit but does not fully meet PLOS ONE’s publication criteria as it currently stands. Therefore, we invite you to submit a revised version of the manuscript that addresses the points raised during the review process.

Dear Dr Counihan,

first of all, I apologize for the very long processing time of you manuscript. We have now obtained comments from two additional reviewers; they were aware that the manuscript was already revised and they also had access to the original (first round) comments. 

Both reviewers clearly see the value of this work. Reviewer 3 makes an important point regarding the usability of your approach beyond the specific case studies and the scientists involved. In order to facilitate the uptake of your approach to a wider audience, perhaps a step-by-step guide or a concise follow-through instruction box could really help. Reviewer 4 has only minor comments and suggests edits to aid readability of the manuscript. Please note that Reviewer 4 has included an attachment. 

We will be pleased to accept this paper once you considered both reviewers comments carefully.

Looking forward to receive the last revision.

Many thanks for your patience

Response: Please see our responses to the reviewer comments below. We appreciate that specific comments were made to improve the readability of the manuscript. Per Reviewer 3’s and your suggestion we have included a series of recommended steps as a table (Table 4) and referenced that recommended steps are included in the abstract and conclusion section. Thank you for the suggestion. 

We look forward to receiving your revised manuscript.

Kind regards,

Stefano Larsen

Academic Editor

PLOS ONE

Journal Requirements:

Reviewers' comments:

Reviewer's Responses to Questions

Comments to the Author

1. If the authors have adequately addressed your comments raised in a previous round of review and you feel that this manuscript is now acceptable for publication, you may indicate that here to bypass the “Comments to the Author” section, enter your conflict of interest statement in the “Confidential to Editor” section, and submit your "Accept" recommendation.

Reviewer #3: (No Response)

Reviewer #4: (No Response)

2. Is the manuscript technically sound, and do the data support the conclusions?

Reviewer #3: Yes

Reviewer #4: Yes

3. Has the statistical analysis been performed appropriately and rigorously?

Reviewer #3: N/A

Reviewer #4: Yes

4. Have the authors made all data underlying the findings in their manuscript fully available?

Reviewer #3: (No Response)

Reviewer #4: Yes

5. Is the manuscript presented in an intelligible fashion and written in standard English?

Reviewer #3: Yes

Reviewer #4: Yes

6. Review Comments to the Author

Reviewer #3: This revised manuscript defines and explores a conceptual model setting a structured framework that aims to identify gaps in monitoring information that could inform the management of fish in large rivers. After reading the manuscript, I still have the general feeling that something is lacking to make a more useful contribution to river and fish managers. As it stands, I feel that the manuscript is the result of trying to synthetize discussions and outcomes from the held workshops. But the manuscript tries to offer more than that. If I understand well, the authors want to provide a model to be followed so that, at the end, the model user comes out with the variable(s) that would need to be measured in order to fill the knowledge gap(s) that we need to improve the management of a river or a species. If the authors want their model to be used by other researchers and managers than those that participated to the workshops and those that work on the four study cases, a clear guidance through the different steps should be provided, just as a manual. Otherwise, the manuscript is just the result of a brainstorming that the authors want to share in a paper. I am NOT saying that the information provided in the manuscript is useless or has no rigor or significance. On the contrary, I think the manuscript brings interesting evidence and synthesis. I am just saying that if the authors want their model to be used, they need to provide clear step-by-step guidance. I think this is more useful than case studies if they want the model to be largely used.

PlosOne is publishing papers on the basis of methodological rigor and high ethical standards. Based on that and since the manuscript is rather conceptual, I think the manuscript can be published in this journal without further changes, so that my above general comment is only a suggestion.

L. 114-116. Something is missing in this sentence?

Response: We have removed the sentence to clarify issues noted by Reviewer 4 (see below).

L.264. add “hierarchically” after “the spatial scales considered are”

Response: Thank you we have implemented your suggestion.

L.512-515. This is repeated in many figures. Consider reducing this to avoid repetitions

Response: Thank you, we have removed this text in the body of the manuscript to avoid redundancy but respectfully suggest that the description needs to be repeated in the figures so that they can be viewed as “stand alone” items.

Resolution of the figures must be improved

Response: Thank you, we have confirmed the resolution of our figures using the PACE tool.

Reviewer #4: In this manuscript the authors used a conceptual model to help management of large river monitoring programs in identifying knowledge gaps.

I believe the authors have addressed very well the comments from the first round of review. For this reason the manuscript story and approach sound solid with a good description for the four cases, highlighting the main potential factors of disturbance and knowledge gaps in large rivers as well as when informations are not enough and more data are necessary.

I have included in the attached pdf some small comments to help the authors to improve the readability, and I believe that the authors would not need any additional comment on the main manuscript concept.

Best regards

Response:

L35 – Thank you for the suggestion but we respectfully suggest that adding this language to the abstract may be more detracting than helpful. We understand your point though. We tried to strike a balance here between succinctness and providing detail.

L48 – Thank you, we have incorporated this suggestion

L54 – Thank you, we have incorporated this suggestion

L114, 131, 152 – We have made edits to paragraphs that encompass these comments to clarify the language about the sequencing of events and who was attending the workshop etc. Thank you for this observation. 

L203 - Thank you, we have incorporated this suggestion

L227 – Thank you, we have enumerated the steps in the sentence to retain the connection of the elements and clarify the sequence.

L306 – Thank you, we are aware that the figure captions are long, but this represents our best attempt at minimizing the text while including enough information to understand the complexity in the figures.

L467 – Thank you, we respectfully suggest that the elements in the sentence need to remain together. However, we have edited the sentence to shorten it, hopefully improve the readability, and clarify the content.

L723 – Thank you, for this comment. We have edited the sentence to improve readability.

7. PLOS authors have the option to publish the peer review history of their article (what does this mean?). If published, this will include your full peer review and any attached files.

Do you want your identity to be public for this peer review? For information about this choice, including consent withdrawal, please see our Privacy Policy.

Reviewer #3: No

Reviewer #4: No

---

## [Editor Report · Decision Letter 2]

25 Jan 2022

PONE-D-21-12444R2Identifying monitoring information needs that support the management of fish in large riversPLOS ONE

Dear Dr. Counihan,

Thank you for submitting your manuscript to PLOS ONE. After careful consideration, we feel that it has merit but does not fully meet PLOS ONE’s publication criteria as it currently stands. Therefore, we invite you to submit a revised version of the manuscript that addresses the points raised during the review process.

We look forward to receiving your revised manuscript.

Kind regards,

Stefano Larsen

Academic Editor

PLOS ONE

Additional Editor Comments: 

Dear Dr. Counihan,

Thanks for your patience during this long review process.

It looks as though you have clarified all reviewers' comments, and we are ready to accept the manuscript for publication.

However, although you mentioned to have checked the resolution of the Figures, they still look rather fuzzy to me, and difficult to read. If I zoom in the figures in the manuscript pdf, they really look out of focus.

Please, rework the figures and submit a clearer version.

I have scored the ms as 'minor revision', but note that higher resolution images need to be included.

Looking forward to hear from you

SL
---

## [Author Response · Author response to Decision Letter 2]

1 Mar 2022

PONE-D-21-12444R1

Identifying monitoring information needs that support the management of fish in large rivers

PLOS ONE

Dear Dr. Counihan,

Thank you for submitting your manuscript to PLOS ONE. After careful consideration, we feel that it has merit but does not fully meet PLOS ONE’s publication criteria as it currently stands. Therefore, we invite you to submit a revised version of the manuscript that addresses the points raised during the review process.

Dear Dr Counihan,

first of all, I apologize for the very long processing time of you manuscript. We have now obtained comments from two additional reviewers; they were aware that the manuscript was already revised and they also had access to the original (first round) comments. 

Both reviewers clearly see the value of this work. Reviewer 3 makes an important point regarding the usability of your approach beyond the specific case studies and the scientists involved. In order to facilitate the uptake of your approach to a wider audience, perhaps a step-by-step guide or a concise follow-through instruction box could really help. Reviewer 4 has only minor comments and suggests edits to aid readability of the manuscript. Please note that Reviewer 4 has included an attachment. 

We will be pleased to accept this paper once you considered both reviewers comments carefully.

Looking forward to receive the last revision.

Many thanks for your patience

Response: Please see our responses to the reviewer comments below. We appreciate that specific comments were made to improve the readability of the manuscript. Per Reviewer 3’s and your suggestion we have included a series of recommended steps as a table (Table 4) and referenced that recommended steps are included in the abstract and conclusion section. Thank you for the suggestion. 

We look forward to receiving your revised manuscript.

Kind regards,

Stefano Larsen

Academic Editor

PLOS ONE

Journal Requirements:

Reviewers' comments:

Reviewer's Responses to Questions

Comments to the Author

1. If the authors have adequately addressed your comments raised in a previous round of review and you feel that this manuscript is now acceptable for publication, you may indicate that here to bypass the “Comments to the Author” section, enter your conflict of interest statement in the “Confidential to Editor” section, and submit your "Accept" recommendation.

Reviewer #3: (No Response)

Reviewer #4: (No Response)

2. Is the manuscript technically sound, and do the data support the conclusions?

Reviewer #3: Yes

Reviewer #4: Yes

3. Has the statistical analysis been performed appropriately and rigorously?

Reviewer #3: N/A

Reviewer #4: Yes

4. Have the authors made all data underlying the findings in their manuscript fully available?

Reviewer #3: (No Response)

Reviewer #4: Yes

5. Is the manuscript presented in an intelligible fashion and written in standard English?

Reviewer #3: Yes

Reviewer #4: Yes

6. Review Comments to the Author

Reviewer #3: This revised manuscript defines and explores a conceptual model setting a structured framework that aims to identify gaps in monitoring information that could inform the management of fish in large rivers. After reading the manuscript, I still have the general feeling that something is lacking to make a more useful contribution to river and fish managers. As it stands, I feel that the manuscript is the result of trying to synthetize discussions and outcomes from the held workshops. But the manuscript tries to offer more than that. If I understand well, the authors want to provide a model to be followed so that, at the end, the model user comes out with the variable(s) that would need to be measured in order to fill the knowledge gap(s) that we need to improve the management of a river or a species. If the authors want their model to be used by other researchers and managers than those that participated to the workshops and those that work on the four study cases, a clear guidance through the different steps should be provided, just as a manual. Otherwise, the manuscript is just the result of a brainstorming that the authors want to share in a paper. I am NOT saying that the information provided in the manuscript is useless or has no rigor or significance. On the contrary, I think the manuscript brings interesting evidence and synthesis. I am just saying that if the authors want their model to be used, they need to provide clear step-by-step guidance. I think this is more useful than case studies if they want the model to be largely used.

PlosOne is publishing papers on the basis of methodological rigor and high ethical standards. Based on that and since the manuscript is rather conceptual, I think the manuscript can be published in this journal without further changes, so that my above general comment is only a suggestion.

L. 114-116. Something is missing in this sentence?

Response: We have removed the sentence to clarify issues noted by Reviewer 4 (see below).

L.264. add “hierarchically” after “the spatial scales considered are”

Response: Thank you we have implemented your suggestion.

L.512-515. This is repeated in many figures. Consider reducing this to avoid repetitions

Response: Thank you, we have removed this text in the body of the manuscript to avoid redundancy but respectfully suggest that the description needs to be repeated in the figures so that they can be viewed as “stand alone” items.

Resolution of the figures must be improved

Response: Thank you, we have confirmed the resolution of our figures using the PACE tool.

Reviewer #4: In this manuscript the authors used a conceptual model to help management of large river monitoring programs in identifying knowledge gaps.

I believe the authors have addressed very well the comments from the first round of review. For this reason the manuscript story and approach sound solid with a good description for the four cases, highlighting the main potential factors of disturbance and knowledge gaps in large rivers as well as when informations are not enough and more data are necessary.

I have included in the attached pdf some small comments to help the authors to improve the readability, and I believe that the authors would not need any additional comment on the main manuscript concept.

Best regards

Response:

L35 – Thank you for the suggestion but we respectfully suggest that adding this language to the abstract may be more detracting than helpful. We understand your point though. We tried to strike a balance here between succinctness and providing detail.

L48 – Thank you, we have incorporated this suggestion

L54 – Thank you, we have incorporated this suggestion

L114, 131, 152 – We have made edits to paragraphs that encompass these comments to clarify the language about the sequencing of events and who was attending the workshop etc. Thank you for this observation. 

L203 - Thank you, we have incorporated this suggestion

L227 – Thank you, we have enumerated the steps in the sentence to retain the connection of the elements and clarify the sequence.

L306 – Thank you, we are aware that the figure captions are long, but this represents our best attempt at minimizing the text while including enough information to understand the complexity in the figures.

L467 – Thank you, we respectfully suggest that the elements in the sentence need to remain together. However, we have edited the sentence to shorten it, hopefully improve the readability, and clarify the content.

L723 – Thank you, for this comment. We have edited the sentence to improve readability.

7. PLOS authors have the option to publish the peer review history of their article (what does this mean?). If published, this will include your full peer review and any attached files.

Do you want your identity to be public for this peer review? For information about this choice, including consent withdrawal, please see our Privacy Policy.

Reviewer #3: No

Reviewer #4: No

---

## [Editor Report · Decision Letter 3]

4 Apr 2022

Identifying monitoring information needs that support the management of fish in large rivers

PONE-D-21-12444R3

Dear Dr. Counihan,

We’re pleased to inform you that your manuscript has been judged scientifically suitable for publication and will be formally accepted for publication once it meets all outstanding technical requirements.

Kind regards,

Stefano Larsen

Academic Editor

PLOS ONE

Additional Editor Comments (optional):

Dear Dr Counihan,

thanks for clarifying our queries regarding the Maps.
---

## [Editor Report · Acceptance letter]

21 Apr 2022

PONE-D-21-12444R3 

Identifying monitoring information needs that support the management of fish in large rivers 

Dear Dr. Counihan:

I'm pleased to inform you that your manuscript has been deemed suitable for publication in PLOS ONE. Congratulations! Your manuscript is now with our production department. 

Kind regards, 

on behalf of

Dr. Stefano Larsen 

Academic Editor

PLOS ONE